# Low-Rank Graph Contrastive Learning for Node Classification

## Abstract

Graph Neural Networks (GNNs) have been widely used to learn node representations and with outstanding performance on various tasks such as node classification. However, noise, which inevitably exists in real-world graph data, would considerably degrade the performance of GNNs revealed by recent studies. In this work, we propose a novel and robust GNN encoder, Low-Rank Graph Contrastive Learning (LR-GCL). Our method performs transductive node classification in two steps. First, a low-rank GCL encoder named LR-GCL is trained by prototypical contrastive learning with low-rank regularization. Next, using the features produced by LR-GCL, a linear transductive classification algorithm is used to classify the unlabeled nodes in the graph. Our LR-GCL is inspired by the low frequency property of the graph data and its labels, and it is also theoretically motivated by our sharp generalization bound for transductive learning. To the best of our knowledge, our theoretical result is among the first to theoretically demonstrate the advantage of low-rank learning in graph contrastive learning supported by strong empirical performance. Extensive experiments on public benchmarks demonstrate the superior performance of LR-GCL and the robustness of the learned node representations. The code of LR-GCL is available at https://anonymous.4open.science/r/LRGCL/.

## 1 Introduction

Graph Neural Networks (GNNs) have become popular tools for node representation learning in recent years (Kipf & Welling, 2017; Bruna et al., 2014; Hamilton et al., 2017; Xu et al., 2019b). Most prevailing GNNs (Kipf & Welling, 2017; Zhu & Koniusz, 2020) leverage the graph structure and obtain the representation of nodes in a graph by utilizing the features of their connected nodes. Benefiting from such propagation mechanism, node representations obtained by GNN encoders have demonstrated superior performance on various downstream tasks such as semi-supervised node classification and node clustering. Although GNNs have achieved great success in node representation learning, many existing GNN approaches do not consider the noise in the input graph. In fact, noise inherently exists in the graph data for many real-world applications Zhu et al. (2024); Zhong et al. (2019). Such noise may be present in node attributes or node labels, which forms two types of noise, attribute noise and label noise. Recent works, such as (Patrini et al., 2017), have evidenced that noisy inputs hurt the generalization capability of neural networks. Moreover, noise in a subset of the graph data can easily propagate through the graph topology to corrupt the remaining nodes in the graph data Dai et al. (2021); Wang et al. (2023; 2024b). Nodes that are corrupted by noise or falsely labeled would adversely affect the representation learning of themselves and their neighbors. While manual data cleaning and labeling could be remedies to the consequence of noise, they are expensive processes and difficult to scale, thus not able to handle almost infinite amount of noisy data online. Therefore, it is crucial to design a robust GNN encoder that could make use of noisy training data while circumventing the adverse effect of noise. In this paper, we propose a novel GCL encoder termed Low-Rank Graph Contrastive Learning (LR-GCL) to improve the robustness and the generalization capabilities of node representations for GNNs.

Prior work has demonstrated that deep neural networks can overfit to noisy data, significantly degrading generalization performance (Zhang et al., 2021). Robust learning methods broadly fall into two categories, which are *loss correction*, which modifies the learning objective to reduce the influence of corrupted labels

or features (Patrini et al., 2017; Goldberger & Ben-Reuven, 2016), and *sample selection*, which attempts to identify and train on clean samples only (Malach & Shalev-Shwartz, 2017; Jiang et al., 2018; Yu et al., 2019; Li et al., 2020; Han et al., 2018). While several methods (Dai et al., 2021; Qian et al., 2022; Zhuang & Al Hasan, 2022) have extended these ideas to graph data, they primarily rely on heuristic assumptions and lack theoretical analyses regarding how to improve the robustness of GNNs to noise in semi-supervised node classification. Our LR-GCL is inspired by the low frequency property of the graph data and its labels, and it is also theoretically motivated by our sharp generalization bound for transductive learning. To the best of our knowledge, our theoretical result is among the first to theoretically demonstrate the advantage of low-rank learning in graph contrastive learning supported by strong empirical performance. Extensive experiments on public benchmarks demonstrate the superior performance of LR-GCL and the robustness of the learned node representations.

Although GNNs are considered low-pass filtering, they implicitly learn the low-frequency information, and the effect of such low-pass filtering is not strong enough to capture the Low-Frequency Property (LFP) in the noisy labels. As illustrated by Figure 2 deferred to Section 5.7 and Figure 3 in Section B.3 of the appendix, which demonstrates the LFP, that is, the majority of the clean label information is contained only in the low-rank part of the observed label. In contrast with existing GNNs, our LR-GCL better captures the LFP in the noisy labels by learning low-rank features. We remark that low-rank learning exhibits superior performance for noisy attributes in (Cheng et al., 2021) through learnable low-rank filters. Moreover, recent works on graph attention/transformer have shown that finding a good balance between low-frequency and high-frequency information in the graph benefits node representation learning for graph learning tasks such as node classification (Choi et al., 2024a; Zhang et al., 2024a). Compared with the existing GNNs and graph attention/transformer methods, our LR-GCL learns a better balance between low-frequency and high-frequency information, with more focus on the low-frequency part by minimizing the Truncated Nuclear Norm (TNN) due to LFP. As shown in the new Table 3 and Table 8 in Section 5.3, LR-GCL exhibits better node classification accuracy than graph attention/transformer methods, GFSA (Choi et al., 2024a) and HONGAT (Zhang et al., 2024a), when label noise or attribute noise is present in the input graph. In addition, the balance between the low-frequency and high-frequency information can be quantitatively measured by the kernel complexity defined in Section 4.2. As shown in Table 5 and Table 6 in Section 5.5, the node representations learned by LR-GCL exhibit lower kernel complexity than those of graph contrastive learning methods and graph attention/transformer methods.

## 1.1 Contributions

Our contributions are as follows.

First, we present a novel and provable GCL encoder termed Low-Rank Graph Contrastive Learning (LR-GCL). Our algorithm is inspired by the low frequency property illustrated in Figure 2. That is, the low-rank projection of the ground truth clean labels possesses the majority of the information of the clean labels, and projection of the label noise is mostly uniform over all the eigenvectors of a kernel matrix used in classification. Inspired by this observation, LR-GCL adds the TNN as a low-rank regularization term in the loss function of the regular prototypical graph contrastive learning. As a result, the features produced by LR-GCL tend to be low-rank, and such low-rank features are the input to the linear transductive classification algorithm. We provide a novel generalization bound for the test loss on the unlabeled data, and our bound is among the first few works which exhibit the advantage of learning with low-rank features for transductive classification with the presence of noise.

Second, we provide strong theoretical guarantee on the generalization capability of the linear transductive algorithm with the low-rank features produced by LR-GCL as the input. Extensive experimental results on popular graph datasets evidence the advantage of LR-GCL over competing methods for node classification on noisy graph data.

The organization of this paper is described as follows. In Section 2, we review existing graph neural networks, graph contrastive learning approaches, and robust learning techniques that motivate our method. Section 3 formally defines the learning objective, the notations, and the assumptions of our node classification task under noisy conditions. In Section 4, we present the formulation of the proposed Low-Rank

Graph Contrastive Learning (LR-GCL) method with theoretical guarantee. Next, Section 5 validates our approach through extensive comparisons across benchmarks under varying noise conditions, demonstrating the superiority of LR-GCL.

## 2 Related Works

### 2.1 Graph Neural Networks

Graph neural networks (GNNs) have recently become popular tools for node representation learning. Given the difference in the convolution domain, current GNNs fall into two classes. The first class features spectral convolution (Bruna et al., 2014; Kipf & Welling, 2017), and the second class (Hamilton et al., 2017; Veličković et al., 2017; Xu et al., 2019b) generates node representations by sampling and propagating features from their neighborhood. To learn node representation without node labels, contrastive learning has recently been applied to the training of GNNs (Suresh et al., 2021; Thakoor et al., 2021; Wang et al., 2022; Lee et al., 2022; Feng et al., 2022a; Zhang et al., 2023; Lin et al., 2023). Most proposed graph contrastive learning methods (Veličković et al., 2019; Sun et al., 2019; Hu et al., 2019; Jiao et al., 2020; Peng et al., 2020; You et al., 2021; Jin et al., 2021; Mo et al., 2022) create multiple views of the unlabeled input graph and maximize agreement between the node representations of these views. For example, SFA (Zhang et al., 2023) manipulates the spectrum of the node embeddings to construct augmented views in graph contrastive learning. In addition to constructing node-wise augmented views, recent works (Xu et al., 2021; Guo et al., 2022; Li et al., 2021) propose to perform contrastive learning between node representations and semantic prototype representations (Snell et al., 2017; Arik & Pfister, 2020; Allen et al., 2019; Xu et al., 2020) to encode the global semantics information.

However, as pointed out by (Dai et al., 2021), the performance of GNNs can be easily degraded by noisy training data (NT et al., 2019). Moreover, the adverse effects of noise in a subset of nodes can be exaggerated by being propagated to the remaining nodes through the network structure, exacerbating the negative impact of noise Wang et al. (2024b). Unlike previous GCL methods, we propose using contrastive learning to train GNN encoders that are robust to noise existing in the labels and attributes of nodes.

### 2.2 Existing Methods Handing Noisy Data

Previous works (Zhang et al., 2021) have shown that deep neural networks usually generalize badly when trained on input with noise. Existing literature on robust learning mostly fall into two categories. The first category (Patrini et al., 2017; Goldberger & Ben-Reuven, 2016) mitigates the effects of noisy inputs by correcting the computation of loss function, known as loss corruption. The second category aims to select clean samples from noisy inputs for the training (Malach & Shalev-Shwartz, 2017; Jiang et al., 2018; Yu et al., 2019; Li et al., 2020; Han et al., 2018), known as sample selection.

To improve the performance of GNNs on graph data with noise, NRGNN(Dai et al., 2021) first introduces a graph edge predictor to predict missing links for connecting unlabeled nodes with labeled nodes. RTGNN (Qian et al., 2022) trains a robust GNN classifier with scarce and noisy node labels. It first classifies labeled nodes into clean and noisy ones and adopts reinforcement supervision to correct noisy labels. To improve the robustness of the node classifier on the dynamic graph, GraphSS (Zhuang & Al Hasan, 2022) proposes to generalize noisy supervision as a kind of self-supervised learning method, which regards the noisy labels, including both manual-annotated labels and auto-generated labels, as one kind of self-information for each node. Different from previous works, we aim to improve the robustness of GNN encoders for node classification by applying low-rank regularization during the training of the transductive classifier.

### 2.3 Learning Low-Frequency Signal in Graphs with GNNs and Graph Attention

Conventional GNNs, such as the Graph Convolutional Network (GCN) (Kipf & Welling, 2017), learn node representations by aggregating information from their neighbors, inherently functioning as low-pass filters. Existing works (NT & Maehara, 2019; Xu et al., 2019a; Wu et al., 2019; Yu & Qin, 2020) suggest that capturing low-frequency information in the graph structure and node features is crucial to the success of GNNs.

However, recent studies (Bo et al., 2021; Zhang et al., 2024b; Dong et al., 2025) indicate that relying solely on low-frequency information can lead to over-smoothing (Sun et al., 2022), potentially degrading GNN performance on graph datasets where nodes from different classes are frequently connected. To address this issue, recent studies (Bo et al., 2021; Dong et al., 2021; Ju et al., 2022) have proposed methods to adaptively balance low-frequency and high-frequency information in learned node representations, demonstrating improvements in graph learning tasks such as node classification (Tang et al., 2025). Furthermore, recent studies have shown that GNNs explicitly designed to emphasize learning on the low-rank components of node features and graph topology can enhance the robustness of GNNs against the noise in the graph (Tang et al., 2024; Yang et al., 2023).

In addition, recent studies have shown that graph attention mechanisms, such as the Graph Attention Network (GAT) (Veličković et al., 2017), can also facilitate the learning of low-frequency information (Zhang et al., 2024a; Choi et al., 2024b). To mitigate over-smoothing in graph attention, HONGAT (Zhang et al., 2024a) enhances correlation learning among high-order neighbors and sparsifies the attention weight matrix. Moreover, recent works have explored the integration of spectral filters with graph attention to achieve a more balanced and adaptive learning of different frequency components in node representations (Chang et al., 2021; Sun et al., 2024; Wang et al., 2024a).

## 3 Problem Setup

### 3.1 Notations

An attributed graph with $N$ nodes is denoted as $\mathcal{G} = (\mathcal{V}, \mathcal{E}, \mathbf{X})$, where the node set $\mathcal{V} = \{v_1, v_2, \ldots, v_N\}$ and the edge set $\mathcal{E} \subseteq \mathcal{V} \times \mathcal{V}$ represent the nodes and edges of the graph, respectively. The matrix $\mathbf{X} \in \mathbb{R}^{N \times D}$ denotes the attributes for all the nodes, where $D$ is the dimension of node attributes. The adjacency matrix $\mathbf{A} \in \{0, 1\}^{N \times N}$ for $\mathcal{G}$ has elements $\mathbf{A}_{ij} = 1$ if there is an edge $(v_i, v_j) \in \mathcal{E}$. When self-loops are added to the graph, the modified adjacency matrix is given by $\tilde{\mathbf{A}} = \mathbf{A} + \mathbf{I}$, and $\tilde{\mathbf{D}}$ is the diagonal degree matrix corresponding to $\tilde{\mathbf{A}}$. The notation $[N]$ denotes all natural numbers from 1 to $N$ inclusive. $\mathcal{L}$ is a subset of $[N]$ of size $m$, and $\mathcal{U} = [N] \setminus \mathcal{L}$ and $|\mathcal{U}| = u$. Let $\mathcal{V}_\mathcal{L}$ and $\mathcal{V}_\mathcal{U}$ denote the set of labeled nodes and unlabeled test nodes, respectively, and $|\mathcal{V}_\mathcal{L}| = m$, $|\mathcal{V}_\mathcal{U}| = u$. Let $\mathbf{u} \in \mathbb{R}^N$ be a vector, we use $[\mathbf{u}]_\mathcal{A}$ to denote a vector formed by elements of $\mathbf{u}$ with indices in $\mathcal{A}$ for $\mathcal{A} \subseteq [N]$. If $\mathbf{u}$ is a matrix, then $[\mathbf{u}]_\mathcal{A}$ denotes a submatrix formed by rows of $\mathbf{u}$ with row indices in $\mathcal{A}$. $\|\cdot\|_\mathrm{F}$ denotes the Frobenius norm of a matrix, and $\|\cdot\|_p$ denotes the $p$-norm of a vector.

### 3.2 Graph Convolution Network (GCN)

To learn the node representation from the attributes $\mathbf{X}$ and the graph structure $\mathbf{A}$, one simple yet effective neural network model is Graph Convolution Network (GCN). GCN is originally proposed for semi-supervised node classification, which consists of two graph convolution layers. In our work, we use GCN as the backbone of the proposed LR-GCL, which is the GCL encoder, to obtain node representation $\hat{\mathbf{H}} \in \mathbb{R}^{N \times d}$, where the $i$-th row of $\hat{\mathbf{H}}$ is the node representation of $v_i$. In this manner, the output of LR-GCL is $\hat{\mathbf{H}} = g(\mathbf{X}, \mathbf{A}) = \sigma(\hat{\mathbf{A}} \sigma(\hat{\mathbf{A}} \mathbf{X} \tilde{\mathbf{W}}^{(0)}) \tilde{\mathbf{W}}^{(1)})$, where $\hat{\mathbf{A}} = \tilde{\mathbf{D}}^{-1/2} \tilde{\mathbf{A}} \tilde{\mathbf{D}}^{-1/2}$, $\tilde{\mathbf{W}}^{(0)}$ and $\tilde{\mathbf{W}}^{(1)}$ are the weight matrices, and $\sigma$ is the activation function ReLU. The robust and low-rank node representations produced by the LR-GCL are used to perform transductive node classification by a linear classifier. LR-GCL and the linear transductive node classification algorithm are detailed in Section 4.

### 3.3 Problem Description

Noise usually exists in the input node attributes or labels of real-world graphs, which degrades the quality of the node representation obtained by common GCL encoders and affects the performance of the classifier trained on such representations. We aim to obtain node representations robust to noise in two cases, where noise is present in either the labels of $\mathcal{V}_\mathcal{L}$ or in the input node attributes $\mathbf{X}$. That is, we consider either noisy label or noisy input node attributes.

The goal of LR-GCL is to learn low-rank node representations by $\mathbf{H} = g(\mathbf{X}, \mathbf{A})$ such that the node representations $\{\mathbf{h}_i\}_{i=1}^N$ are robust to noise in the above two cases, where $g(\cdot)$ is the LR-GCL encoder. In our work, $g$ is a two-layer GCN introduced in Section 3.2. The low-rank node representations by LR-GCL, $\mathbf{H} = \{\mathbf{h}_1; \mathbf{h}_2; \ldots; \mathbf{h}_N\} \in \mathbb{R}^{N \times d}$, are used for transductive node classification by a linear classifier. In transductive node classification, a linear transductive classifier is trained on $\mathcal{V}_{\mathcal{L}}$, and then the classifier predicts the labels of the unlabeled test nodes in $\mathcal{V}_{\mathcal{U}}$.

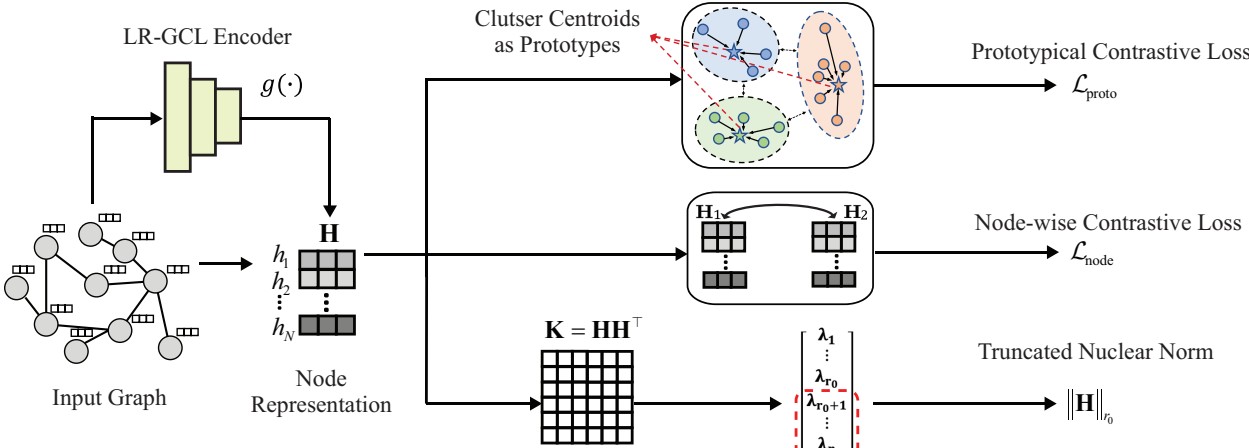

Figure 1: Illustration of the LR-GCL framework.

# 4 Methods

## 4.1 Low-Rank GCL: Low-Rank Graph Contrastive Learning

**Preliminary of Prototypical GCL.** The general node representation learning aims to train an encoder $g(\cdot)$, which is a two-layer Graph Convolution Neural Network (GCN) (Kipf & Welling, 2017), to generate discriminative node representations. In our work, we adopt contrastive learning to train the GCL encoder $g(\cdot)$. To perform contrastive learning, two different views, $G^1 = (\mathbf{X}^1, \mathbf{A}^1)$ and $G^2 = (\mathbf{X}^2, \mathbf{A}^2)$, are generated by node dropping, edge perturbation, and attribute masking. The representation of two generated views are denoted as $\mathbf{H}^1 = g(\mathbf{X}^1, \mathbf{A}^1)$ and $\mathbf{H}^2 = g(\mathbf{X}^2, \mathbf{A}^2)$, with $\mathbf{H}_i^1$ and $\mathbf{H}_i^2$ being the $i$-th row of $\mathbf{H}^1$ and $\mathbf{H}^2$, respectively. It is preferred that the mutual information between $\mathbf{H}^1$ and $\mathbf{H}^2$ is maximized. For computational reason, its lower bound is usually used as the objective for contrastive learning. We use InfoNCE (Li et al., 2021) as our node-wise contrastive loss. In addition to the node-wise contrastive learning, we also adopt prototypical contrastive learning (Li et al., 2021) to capture semantic information in the node representations, which is interpreted as maximizing the mutual information between node representation and a set of estimated cluster prototypes $\{\mathbf{c}_1, ..., \mathbf{c}_K\}$. Following (Li et al., 2021; Snell et al., 2017), we use $K$-means to cluster the node representations $\{\mathbf{h}_i\}_{i=1}^n$ into $K$ clusters and take the clustering centroid of the $k$-th cluster as the $k$-th prototype $\mathbf{c}_k = \frac{1}{|S_k|} \sum_{\mathbf{h}_i \in S_k} \mathbf{h}_i$ for all $k \in [K]$. The loss function of Prototypical GCL is comprised of two terms, $\mathcal{L}_{node}$, the loss function for node-wise contrastive learning, and $\mathcal{L}_{proto}$, the prototypical contrastive learning loss, which are presented below:

$$\mathcal{L}_{\text{node}} = -\frac{1}{N} \sum_{i=1}^N \log \frac{s(\mathbf{H}_i^1, \mathbf{H}_i^2)}{s(\mathbf{H}_i^1, \mathbf{H}_i^2) + \sum_{j=1}^N s(\mathbf{H}_i^1, \mathbf{H}_j^2)}, \quad \mathcal{L}_{\text{proto}} = -\frac{1}{N} \sum_{i=1}^N \log \frac{\exp(\mathbf{H}_i \cdot \mathbf{c}_k / \tau)}{\sum_{k=1}^K \exp(\mathbf{H}_i \cdot \mathbf{c}_k / \tau)}. \quad (1)$$

Here $s(\mathbf{H}_i^1, \mathbf{H}_i^2)$ is the cosine similarity between two node representations, $\mathbf{H}_i^1$ and $\mathbf{H}_i^2$. The node-wise contrastive loss encourages consistency between node representations across two perturbed views of the input graph. This design is particularly helpful in mitigating the impact of attribute noise, as the perturbations simulate different noise patterns. By maximizing agreement between representations from these views, the model learns to extract noise-invariant features that are robust to corruptions in input attributes. The prototypical contrastive loss clusters node representations and enforces alignment between individual nodes

---

**Algorithm 1** Low-Rank Graph Contrastive Learning (LR-GCL)

---

**Input:** The input attribute matrix $\mathbf{X}$, adjacency matrix $\mathbf{A}$, and the training epochs $t_{\max}$.
**Output:** The parameters of LR-GCL encoder $g$
 1: Initialize the parameter of LR-GCL encoder $g$
 2: **for** $t \leftarrow 1$ to $t_{\max}$ **do**
 3:    Calculate node representations by $\mathbf{H} = g(\mathbf{X}, \mathbf{A})$, generate augmented views $G^1, G^2$, and calculate node representations $\mathbf{H}^1 = g(\mathbf{X}^1, \mathbf{A}^1)$ and $\mathbf{H}^2 = g(\mathbf{X}^2, \mathbf{A}^2)$
 4:    Cluster node representations $\{\mathbf{h}_i\}_{i=1}^n$ into $K$ clusters $\{S_k\}_{k=1}^K$ with $K$-means clustering
 5:    Update the prototype $\mathbf{c}_k$ as the centroid of $S_k$ by $\mathbf{c}_k = \frac{1}{|S_k|} \sum_{\mathbf{h}_i \in S_k} \mathbf{h}_i$ for all $k \in [K]$
 6:    Calculate the eigenvalues $\{\lambda_i\}_{i=1}^N$ of the feature kernel $\mathbf{H}^\top \mathbf{H}$
 7:    Update the parameters of LR-GCL encoder $g$ by one step of gradient descent on the loss $\mathcal{L}_{rep}$
 8: **end for**
 9: **return** The LR-GCL encoder $g$

---

and their corresponding cluster prototypes. This helps address label noise by leveraging semantic consistency across nodes within the same cluster. Even if a node's label is corrupted, the prototype, which is computed from a group of similar nodes in a cluster, provides a denoised supervisory signal that guides the representation toward its correct semantic class.

**LR-GCL: Low-Rank Graph Contrastive Learning.** LR-GCL aims to improve the robustness and generalization capability of the node representations of Prototypical GCL by enforcing the learned feature kernel to be low-rank. The kernel gram matrix $\mathbf{K}$ of the node representations $\mathbf{H} \in \mathbb{R}^{N \times d}$ is calculated by $\mathbf{K} = \mathbf{H}^\top \mathbf{H} \in \mathbb{R}^{N \times N}$. Let $\left\{ \widehat{\lambda}_i \right\}_{i=1}^n$ with $\widehat{\lambda}_1 \geq \widehat{\lambda}_2 \ldots \geq \widehat{\lambda}_{\min\{N,d\}} \geq \widehat{\lambda}_{\min\{N,d\}+1} = \ldots, = 0$ be the eigenvalues of $\mathbf{K}$. In order to encourage the features $\mathbf{H}$ or the gram matrix $\mathbf{H}^\top \mathbf{H}$ to be low-rank, we explicitly add the TNN $\|\mathbf{K}\|_{r_0+1} \coloneqq \sum_{r=r_0}^n \widehat{\lambda}_i$ to the loss function of prototypical GCL. The starting rank $r_0 < \min(n, d)$ is the rank of the kernel gram matrix of the features we aim to obtain with the LR-GCL encoder, that is, if $\|\mathbf{K}\|_{r_0} = 0$, then $\text{rank}(\mathbf{K}) = r_0$. Therefore, the overall loss function of LR-GCL is

$$\mathcal{L}_{\text{LR-GCL}} = \mathcal{L}_{\text{node}} + \mathcal{L}_{\text{proto}} + \tau \|\mathbf{K}\|_{r_0}, \tag{2}$$

where $\tau > 0$ is the weighting parameter for the TNN $\|\mathbf{K}\|_{r_0}$. We summarize the training algorithm for the LR-GCL encoder in Algorithm 1. After finishing the training, we calculate the low-rank node feature by $\mathbf{H} = g(\mathbf{A}, \mathbf{X})$.

**Motivation of Learning Low-Rank Features.** Let $\tilde{\mathbf{Y}} \in \mathbb{R}^{N \times C}$ be the ground truth clean label matrix without noise. By the low frequency property illustrated in Figure 2, the projection of $\tilde{\mathbf{Y}}$ on the top $r$ eigenvectors of $\mathbf{K}$ with a small rank $r$, such as $r = 0.2N$, covers the majority of the information in $\tilde{\mathbf{Y}}$. On the other hand, the projection of the label noise $\mathbf{N}$ are distributed mostly uniform across all the eigenvectors. This observation motivates low-rank features $\mathbf{H}$ or equivalently, the low-rank gram matrix $\mathbf{K}$. This is because the low-rank part of the feature matrix $\mathbf{H}$ or the gram matrix $\mathbf{K}$ covers the dominant information in the ground truth label $\tilde{\mathbf{Y}}$ while learning only a small portion of the label noise. Moreover, we remark that the regularization term $\|\mathbf{K}\|_{r_0}$ in the loss function (2) of LR-GCL is also theoretically motivated by the sharp upper bound for the test loss using a linear transductive classifier, presented as (4) in Theorem 4.1. A smaller $\|\mathbf{K}\|_{r_0}$ renders a smaller upper bound for the test loss, which ensures better generalization capability of the linear transductive classier to be introduced in the next subsection.

## 4.2 Transductive Node Classification

In this section, we introduce a simple yet standard linear transductive node classification algorithm using the low-rank node representations $\mathbf{H} \in \mathbb{R}^{N \times d}$ produced by the LR-GCL encoder. We present strong theoretical result on the generalization bound for the test loss for our low-rank transductive algorithm with the presence of label noise.

We first give basic notations for our algorithm. Let $\mathbf{y}_i \in \mathbb{R}^C$ be the observed one-hot class label vector for node $v_i$ for all $i \in [N]$, and define $\mathbf{Y} \coloneqq [\mathbf{y}_1; \mathbf{y}_2; \ldots \mathbf{y}_N] \in \mathbb{R}^{N \times C}$ be the observed label matrix which may

contain label noise $\mathbf{N} \in \mathbb{R}^{N \times C}$. We define $\mathbf{F}(\mathbf{W}) = \mathbf{HW}$ as the linear output of the transductive classier with $\mathbf{W} \in \mathbb{R}^{d \times C}$ being the weight matrix for the classifier. Our transductive classifier uses softmax$(\mathbf{F}(\mathbf{W})) \in \mathbb{R}^{N \times C}$ for prediction of the labels of the test nodes. We train the transductive classifier by minimizing the regular cross-entropy on the labeled nodes through

$$\min_{\mathbf{W}} L(\mathbf{W}) = \frac{1}{m} \sum_{v_i \in \mathcal{V}_{\mathcal{L}}} \mathrm{KL}\left(\mathbf{y}_i, [\mathrm{softmax}\left(\mathbf{HW}\right)]_i\right), \tag{3}$$

where KL is the KL divergence between the label $\mathbf{y}_i$ and the softmax of the classifier output at node $v_i$. We use a regular gradient descent to optimize (3) with a learning rate $\eta \in (0, \frac{1}{\hat{\lambda}_1})$. $\mathbf{W}$ is initialized by $\mathbf{W}^{(0)} = \mathbf{0}$, and at the $t$-th iteration of gradient descent for $t \geq 1$, $\mathbf{W}$ is updated by $\mathbf{W}^{(t)} = \mathbf{W}^{(t-1)} - \eta \nabla_{\mathbf{W}} L(\mathbf{W})|_{\mathbf{W} = \mathbf{W}^{(t-1)}}$.

Define $\mathbf{F}(\mathbf{W}, t) := \mathbf{HW}^{(t)}$ as the output of the classifier after the $t$-th iteration of gradient descent for $t \geq 1$. We have the following theoretical result, Theorem 4.1, on the Mean Squared Error (MSE) loss of the unlabeled test nodes $\mathcal{V}_{\mathcal{U}}$ measured by the gap between $[\mathbf{F}(\mathbf{W}, t)]_{\mathcal{U}}$ and $[\tilde{\mathbf{Y}}]_{\mathcal{U}}$ when using the low-rank feature $\mathbf{H}$ with $r_0 \in [n]$, which is the generalization error bound for the linear transductive classifier using $\mathbf{F}(\mathbf{W}) = \mathbf{HW}$ to predict the labels of the unlabeled nodes. Similar to existing works (Kothapalli et al., 2023) that uses the Mean Squared Error (MSE) to analyze the optimization and the generalization of GNNs, we employ the MSE loss to provide the generalization error of the node classifier in the following theorem. It is remarked that the MSE loss is necessary for the generalization analysis of transductive learning using transductive local Rademacher complexity (Tolstikhin et al., 2014; Yang, 2023).

**Theorem 4.1.** Let $m \geq cN$ for a constant $c \in (0, 1)$, and $r_0 \in [n]$. Assume that a set $\mathcal{L}$ with $|\mathcal{L}| = m$ is sampled uniformly without replacement from $[N]$, and the remaining nodes $\mathcal{V}_U = \mathcal{V} \setminus \mathcal{V}_L$ are the test nodes. Then for every $x > 0$, with probability at least $1 - \exp(-x)$, after the $t$-th iteration of gradient descent for all $t \geq 1$, we have

$$\begin{aligned}
\mathcal{U}_{\mathrm{test}}(t) &:= \frac{1}{u} \big\| [\mathbf{F}(\mathbf{W}, t) - \tilde{\mathbf{Y}}]_{\mathcal{U}} \big\|_{\mathrm{F}}^2 \\
&\leq \frac{2c_0}{m} \left( L_1(\mathbf{K}, \tilde{\mathbf{Y}}, t) + L_2(\mathbf{K}, \mathbf{N}, t) \right) + c_0 \mathrm{KC}(\mathbf{K}) + \frac{c_0 x}{u},
\end{aligned} \tag{4}$$

where $c_0$ is a positive number depending on $\mathbf{U}$, $\left\{ \hat{\lambda}_i \right\}_{i=1}^{r_0}$, and $\tau_0$ with $\tau_0^2 = \max_{i \in [N]} \mathbf{K}_{ii}$. $L_1(\mathbf{K}, \tilde{\mathbf{Y}}, t) := \left\| \left( \mathbf{I}_m - \eta \left[ \mathbf{K} \right]_{\mathcal{L}, \mathcal{L}} \right)^t [\tilde{\mathbf{Y}}]_{\mathcal{L}} \right\|_{\mathrm{F}}^2$, $L_2(\mathbf{K}, \mathbf{N}, t) = \left\| \eta \left[ \mathbf{K} \right]_{\mathcal{L}, \mathcal{L}} \sum_{t'=0}^{t-1} \left( \mathbf{I}_m - \eta \left[ \mathbf{K} \right]_{\mathcal{L}, \mathcal{L}} \right)^{t'} [\mathbf{N}]_{\mathcal{L}} \right\|_{\mathrm{F}}^2$. KC is the kernel complexity of the kernel gram matrix $\mathbf{K} = \mathbf{HH}^\top$ defined by

$$\mathrm{KC}(\mathbf{K}) = \min_{r_0 \in [0, n]} r_0 \left( \frac{1}{u} + \frac{1}{m} \right) + \sqrt{\|\mathbf{K}\|_{r_0}} \left( \frac{1}{\sqrt{u}} + \frac{1}{\sqrt{m}} \right). \tag{5}$$

This theorem is proved in Section A of the appendix. It is noted that $\mathcal{U}_{\mathrm{test}}(t)$ is the test loss of the unlabeled nodes measured by the distance between the classifier output $\mathbf{F}(\mathbf{W}, t)$ and $\tilde{\mathbf{Y}}$. There are three terms on the upper bound for the test loss in (4), $L_1(\mathbf{K}, \tilde{\mathbf{Y}}, t)$, $L_2(\mathbf{K}, \mathbf{N}, t)$, and $\mathrm{KC}(\mathbf{K})$, which are explained as follows. $L_1(\mathbf{K}, \tilde{\mathbf{Y}}, t)$ corresponds to the training loss of the node classifier with the clean label. $L_2(\mathbf{K}, \mathbf{N}, t)$ corresponds to the loss incurred by label noise. $\mathrm{KC}(\mathbf{K})$ is the kernel complexity (KC), which measures the complexity of the kernel gram matrix from the node representation $\mathbf{H}$ generated by our LR-GCL encoder. We remark that the TNN $\|\mathbf{K}\|_{r_0}$ appears on the RHS of the upper bound (4), theoretically justifying why we learn the low-rank features $\mathbf{K}$ of the LR-GCL by adding the TNN $\|\mathbf{K}\|_{r_0}$ to the loss of our LR-GCL in (2). Moreover, when the low frequency property holds, which is always the case as demonstrated by Figure 2 and Figure 3 in the appendix, $L_1(\mathbf{K}, \tilde{\mathbf{Y}}, t)$ would be very small with enough iteration number $t$. $L_2(\mathbf{K}, \mathbf{N}, t)$ is also small due to the fact that the projection of label noise is approximately uniform over all the eigenvectors, and $\mathbf{K} = \mathbf{H}^\top \mathbf{H}$ is approximately a low-rank matrix of rank $r_0$ since $\mathbf{H}$ is approximately a rank-$r_0$ matrix with its TNN optimized through the optimization of the LR-GCL encoder (2).

In our empirical study in the next section, we search for the rank $r_0$ for the TNN by standard cross-validation for all the graph data sets. In Table 1 of our experimental results, it is observed that the best rank $r_0$ is always between $0.1 \min\{N, d\}$ and $0.3 \min\{N, d\}$. The overall framework of LR-GCL is illustrated in Figure 1.

### 4.3 LRA-LR-GCL: Improving LR-GCL by Low Rank Attention

To further improve the performance of LR-GCL, we introduce LRA-LR-GCL in this section. LRA-LR-GCL features a novel LR-Attention layer, or the LRA layer, which applies self-attention to the output of the LR-GCL encoder by $\mathbf{F} = \mathbf{B}\mathbf{H}$, where $\mathbf{H} \in \mathbb{R}^{N \times d}$ is the low-rank node representations produced by the LR-GCL encoder through the optimization of (2). $\mathbf{F}$ is the attention output and $\mathbf{B} \in \mathbb{R}^{N \times N}$ is our new attention matrix in the LRA layer. We recall that the kernel gram matrix of the node features $\mathbf{H}$ is $\mathbf{K} = \mathbf{H}\mathbf{H}^\top$. The attention weight matrix $\mathbf{B}$ is set to $\mathbf{B} = \mathbf{K}/\widehat{\lambda}_1$. The gram matrix $\mathbf{K_F}$ of the node representations $\mathbf{F} \in \mathbb{R}^{N \times d}$ is then $\mathbf{K_F} = \mathbf{F}\mathbf{F}^\top = \mathbf{K}^3/\widehat{\lambda}_1^2$. Let $\{\lambda_i\}_{i=1}^N$ be the eigenvalues of $\mathbf{K_F}$ with $\lambda_1 \geq \lambda_2 \geq ... \lambda_N \geq 0$, then we have $\lambda_i = \widehat{\lambda}_i^3/\widehat{\lambda}_1^2$ for every $i \in [n]$. Noting that $\lambda_i = \widehat{\lambda}_i \cdot \widehat{\lambda}_i^2/\widehat{\lambda}_1^2 \leq \widehat{\lambda}_i$ due to $\lambda_1 \geq \lambda_i$ for all $i \in [N]$, therefore, the LRA layer can reduce the kernel complexity of the kernel gram matrix $\mathbf{K}$, because the KC of $\mathbf{K_F}$ is always not greater than that of $\mathbf{K}$. We then train a transductive classifier on top of $\mathbf{F}$ similar to Section 4.2 by minimizing the loss function

$$\min_{\mathbf{W}} L(\mathbf{W}) = \frac{1}{m} \sum_{v_i \in \mathcal{V}_\mathcal{L}} \mathrm{KL}\left(\mathbf{y}_i, [\mathrm{softmax}\left(\mathbf{F}\mathbf{W}\right)]_i\right), \tag{6}$$

where $\mathbf{W}$ is the weight matrix for the classifier. Such linear classifier trained with the the LRA layer through the optimization of (6) is termed LRA-LR-GCL. It then follows from the above discussion and the upper bound for the test loss (4) in Theorem 4.1 that LRA-LR-GCL has a lower KC, so that the test loss $\mathcal{U}_{\mathrm{test}}(t)$ of LRA-LR-GCL can be even lower than that of LR-GCL, suggesting a better prediction accuracy of LRA-LR-GCL than LR-GCL. This is empirically justified in Table 5 and Table 6 where LRA-LR-GCL exhibits lower KC and lower upper bound for the test loss than that of LR-GCL.

## 5 Experiments

In this section, we evaluate the performance of LR-GCL on public graph datasets. In Section 5.1, we discuss the experimental settings and implementation details of LRA-GCL. The detailed statistics of the benchmark datasets are presented in Section 5.2. In Section 5.3, we present evaluation results of LR-GCL for semi-supervised node classification with different types of noise. In Section 5.4, we compare LR-GCL with existing GCL methods equipped with different types of classifiers. In Section 5.5, we study the kernel complexity of node representations learned by LR-GCL. In Section 5.6, we perform an ablation study on the rank $r_0$ in the TNN. In Section B.1 of the appendix, we present experiment results for node classification on additional benchmarks. In Section B.2 of the appendix, we compare the training time of LR-GCL with other baseline methods. Additional eigen-projection and signal concentration ratio results are presented in Section B.3 of the appendix. In Section B.4, we study the effectiveness of LR-GCL on the heterophilic graph datasets.

### 5.1 Experimental Settings

In our experiment, we adopt eight widely used graph benchmark datasets, namely Cora, Citeseer, PubMed (Sen et al., 2008), Coauthor CS, ogbn-arxiv (Hu et al., 2020), Wiki-CS (Mernyei & Cangea, 2020), Amazon-Computers, and Amazon-Photos (Shchur et al., 2018) for the evaluation in node classification. Due to the fact that all public benchmark graph datasets do not come with corrupted labels or attribute noise, we manually inject noise into public datasets to evaluate our algorithm. We follow the commonly used label noise generation methods from the existing work (Han et al., 2020; Dai et al., 2022; Qian et al., 2022) to inject label noise. We generate noisy labels over all classes in two types: (1) Symmetric, where nodes from each class is flipped to other classes with a uniform random probability; (2) Asymmetric, where mislabeling only occurs between similar classes. In this work, we adopt the formal definitions of label noise introduced in (Song et al., 2022). Let $\mathbf{T} \in [0, 1]^{C \times C}$ denote the noise transition matrix, where $\mathbf{T}_{ij} := \mathbb{P}(\tilde{y} = j \mid y = i)$ represents the probability that a clean label $y = i$ is flipped to a noisy label $\tilde{y} = j$. Under symmetric noise with rate $\tau \in [0, 1]$, labels are flipped uniformly to any of the other classes, i.e., $\mathbf{T}_{ii} = 1 - \tau$ and $\mathbf{T}_{ij} = \frac{\tau}{C-1}$ for all $j \neq i$. In contrast, asymmetric noise assumes that mislabeling is biased toward specific confounding classes. Formally, $\mathbf{T}_{ii} = 1 - \tau$ and there exist $j \neq i$, $k \neq i$ such that $\mathbf{T}_{ij} > \mathbf{T}_{ik}$, meaning that some incorrect

classes are more likely than others. This setting captures more realistic scenarios where label confusion follows a structured pattern.

To evaluate the performance of our method with attribute noise, we randomly shuffle a certain percentage of input attributes for each node following (Ding et al., 2022). The percentage of shuffled attributes is defined as the attribute noise level in our experiments.

Details on the datasets we use in our experiments are introduced in Section 5.2. For all our experiments, we follow the default separation (Shchur et al., 2018; Mernyei & Cangea, 2020; Hu et al., 2020) of training, validation, and test sets on each benchmark. The noise is added to the training and validation sets, and the test set is kept clean for evaluation. We search for the optimal values of different hyper-parameters, including learning rate, weight decay, hidden dimension, and dropout rate, by 5-fold cross-validation on the training data of each dataset. We search for the learning rate from $\{1 \times 10^{-4}, 5 \times 10^{-4}, 1 \times 10^{-3}, 5 \times 10^{-3}, 1 \times 10^{-2}, 3 \times 10^{-2}, 6 \times 10^{-2}, 1 \times 10^{-1}, 5 \times 10^{-1}\}$. We search for weight decay from $\{1 \times 10^{-5}, 5 \times 10^{-5}, 1 \times 10^{-4}, 5 \times 10^{-4}, 1 \times 10^{-3}, 5 \times 10^{-3}\}$. The dropout rate is selected from $\{0.3, 0.4, 0.5, 0.6, 0.7\}$. Values leading to the lowest validation loss are selected for each dataset. All models are trained using the Adam optimizer for a maximum of 500 epochs, with early stopping applied if the validation loss does not decrease for 20 consecutive epochs. To mitigate the impact of the randomness, we run each experiment for 10 times with different random seeds for the initialization of the network parameters.

**Tuning $r_0, \tau$ by Cross-Validation.** We tune the rank $r_0$ and the weight for the truncated nuclear loss $\tau$ by standard cross-validation on each dataset. Let $r_0 = \lceil \gamma \min \{N, d\} \rceil$ where $\gamma$ is the rank ratio. We select the values of $\gamma$ and $\tau$ by performing 5-fold cross-validation on 20% of the training data in each dataset. The value of $\gamma$ is selected from $\{0.1, 0.2, 0.3, 0.4, 0.5, 0.6, 0.7, 0.8, 0.9\}$. The value of $\tau$ is selected from $\{0.05, 0.1, 0.15, 0.2, 0.25, 0.3, 0.35, 0.4, 0.45, 0.5\}$. The selected values on each dataset are shown in Table 1.

Table 1: Selected rank ratio $\gamma$ and truncated nuclear loss's weight $\lambda$ for each dataset.

| Hyper-parameters | Cora | Citeseer | PubMed | Coauthor CS | ogbn-arxiv | Wiki-CS | Amazon-Computers | Amazon-Photos |
|---|---|---|---|---|---|---|---|---|
| $\tau$ | 0.10 | 0.10 | 0.10 | 0.20 | 0.10 | 0.25 | 0.20 | 0.20 |
| $\gamma$ | 0.2 | 0.2 | 0.3 | 0.3 | 0.4 | 0.2 | 0.2 | 0.3 |

## 5.2 Datasets

We evaluate our method on eight public benchmarks that are widely used for node representation learning, namely Cora, Citeseer, PubMed (Sen et al., 2008), Coauthor CS, ogbn-arxiv (Hu et al., 2020), Wiki-CS (Mernyei & Cangea, 2020), Amazon-Computers, and Amazon-Photos (Shchur et al., 2018). Cora, Citeseer, and PubMed are the three most widely used citation networks. Coauthor CS is a co-authorship graph. The ogbn-arxiv is a directed citation graph. Wiki-CS is a hyperlink networks of computer science articles.

Table 2: The statistics of the datasets.

| Dataset | Nodes | Edges | Features | Classes |
|---|---|---|---|---|
| **Cora** | 2,708 | 5,429 | 1,433 | 7 |
| **CiteSeer** | 3,327 | 4,732 | 3,703 | 6 |
| **PubMed** | 19,717 | 44,338 | 500 | 3 |
| **Coauthor CS** | 18,333 | 81,894 | 6,805 | 15 |
| **ogbn-arxiv** | 169,343 | 1,166,243 | 128 | 40 |
| **Wiki-CS** | 11,701 | 215,863 | 300 | 10 |
| **Amazon-Computers** | 13,752 | 245,861 | 767 | 10 |
| **Amazon-Photos** | 7,650 | 119,081 | 745 | 8 |

Amazon-Computers and Amazon-Photos are co-purchase networks of products selling on Amazon.com. We summarize the statistics of all the datasets in Table 2.

## 5.3 Node Classification

**Compared Methods.** We compare LR-GCL against semi-supervised node representation learning methods, GCN (Kipf & Welling, 2017), GCE (Zhang & Sabuncu, 2018), S$^2$GC (Zhu & Koniusz, 2020), and GRAND+ (Feng et al., 2022b). Furthermore, we include two baseline methods for node classification with label noise, which are NRGNN (Dai et al., 2021) and RTGNN (Qian et al., 2022). We also compare LR-GCL against state-of-the-art GCL methods, including GraphCL (You et al., 2020), MERIT (Jin et al., 2021),

Table 3: Performance comparison for node classification on Cora, Citeseer, PubMed, and Wiki-CS with asymmetric label noise, symmetric label noise, and attribute noise.

| Dataset | Methods | Noise Type | | | | | | | | | |
|---|---|---|---|---|---|---|---|---|---|---|---|
| | | 0 | 40 | | | 60 | | | 80 | | |
| | | - | Asymmetric | Symmetric | Attribute | Asymmetric | Symmetric | Attribute | Asymmetric | Symmetric | Attribute |
| Cora | GCN | 0.815±0.005 | 0.547±0.015 | 0.636±0.007 | 0.639±0.008 | 0.405±0.014 | 0.517±0.010 | 0.439±0.012 | 0.265±0.012 | 0.354±0.014 | 0.317±0.013 |
| | S²GC | 0.835±0.002 | 0.569±0.007 | 0.664±0.007 | 0.661±0.007 | 0.422±0.010 | 0.535±0.010 | 0.454±0.011 | 0.279±0.014 | 0.366±0.014 | 0.320±0.013 |
| | GCE | 0.819±0.004 | 0.573±0.011 | 0.652±0.008 | 0.650±0.014 | 0.449±0.011 | 0.509±0.011 | 0.445±0.015 | 0.280±0.013 | 0.353±0.013 | 0.325±0.015 |
| | UnionNET | 0.820±0.006 | 0.569±0.014 | 0.664±0.007 | 0.653±0.012 | 0.452±0.010 | 0.541±0.010 | 0.450±0.009 | 0.283±0.014 | 0.370±0.011 | 0.320±0.012 |
| | NRGNN | 0.822±0.006 | 0.571±0.019 | 0.676±0.007 | 0.645±0.012 | 0.470±0.014 | 0.548±0.014 | 0.451±0.011 | 0.282±0.022 | 0.373±0.012 | 0.326±0.010 |
| | RTGNN | 0.828±0.003 | 0.570±0.010 | 0.682±0.008 | 0.678±0.011 | 0.474±0.011 | 0.555±0.010 | 0.457±0.009 | 0.280±0.011 | 0.386±0.014 | 0.342±0.016 |
| | SUGRL | 0.834±0.005 | 0.564±0.011 | 0.674±0.012 | 0.675±0.009 | 0.468±0.011 | 0.552±0.011 | 0.452±0.012 | 0.280±0.012 | 0.381±0.012 | 0.338±0.014 |
| | MERIT | 0.831±0.005 | 0.560±0.008 | 0.670±0.008 | 0.671±0.009 | 0.467±0.013 | 0.547±0.013 | 0.450±0.014 | 0.277±0.013 | 0.385±0.013 | 0.335±0.009 |
| | ARIEL | 0.843±0.004 | 0.573±0.013 | 0.681±0.010 | 0.675±0.009 | 0.471±0.012 | 0.553±0.012 | 0.455±0.014 | 0.284±0.014 | 0.389±0.013 | 0.343±0.013 |
| | SFA | 0.839±0.010 | 0.564±0.011 | 0.677±0.013 | 0.676±0.015 | 0.473±0.014 | 0.549±0.014 | 0.457±0.014 | 0.282±0.016 | 0.389±0.013 | 0.344±0.017 |
| | Sel-Cl | 0.828±0.002 | 0.570±0.010 | 0.685±0.012 | 0.676±0.009 | 0.472±0.013 | 0.554±0.014 | 0.455±0.011 | 0.282±0.017 | 0.389±0.013 | 0.341±0.015 |
| | Jo-SRC | 0.825±0.005 | 0.571±0.006 | 0.684±0.013 | 0.679±0.007 | 0.473±0.011 | 0.556±0.008 | 0.458±0.012 | 0.285±0.013 | 0.387±0.018 | 0.345±0.018 |
| | GRAND+ | 0.858±0.006 | 0.570±0.009 | 0.682±0.007 | 0.678±0.011 | 0.472±0.010 | 0.554±0.008 | 0.456±0.012 | 0.284±0.015 | 0.387±0.015 | 0.345±0.013 |
| | GFSA | 0.837±0.006 | 0.568±0.012 | 0.676±0.010 | 0.672±0.009 | 0.466±0.012 | 0.545±0.013 | 0.451±0.012 | 0.279±0.012 | 0.384±0.015 | 0.336±0.013 |
| | HONGAT | 0.833±0.004 | 0.566±0.011 | 0.673±0.011 | 0.667±0.010 | 0.464±0.010 | 0.543±0.011 | 0.449±0.010 | 0.278±0.013 | 0.381±0.014 | 0.334±0.014 |
| | CRGNN | 0.842±0.005 | 0.572±0.010 | 0.678±0.010 | 0.674±0.010 | 0.470±0.012 | 0.551±0.013 | 0.454±0.013 | 0.283±0.014 | 0.386±0.014 | 0.341±0.015 |
| | CGNN | 0.835±0.006 | 0.567±0.009 | 0.670±0.012 | 0.669±0.011 | 0.462±0.013 | 0.544±0.011 | 0.450±0.013 | 0.281±0.012 | 0.380±0.013 | 0.337±0.014 |
| | LR-GCL | 0.858±0.006 | 0.589±0.011 | 0.713±0.007 | 0.695±0.011 | 0.492±0.011 | 0.587±0.013 | 0.477±0.012 | 0.306±0.012 | 0.419±0.012 | 0.363±0.011 |
| | LRA-LR-GCL | **0.861±0.006** | **0.602±0.011** | **0.724±0.007** | **0.708±0.011** | **0.510±0.011** | **0.605±0.013** | **0.492±0.012** | **0.329±0.012** | **0.436±0.012** | **0.382±0.011** |
| Citeseer | GCN | 0.703±0.005 | 0.475±0.023 | 0.501±0.013 | 0.529±0.009 | 0.351±0.014 | 0.341±0.014 | 0.372±0.011 | 0.291±0.022 | 0.281±0.019 | 0.290±0.014 |
| | S²GC | 0.736±0.005 | 0.488±0.013 | 0.528±0.013 | 0.553±0.008 | 0.363±0.012 | 0.367±0.014 | 0.390±0.013 | 0.304±0.024 | 0.284±0.019 | 0.288±0.011 |
| | GCE | 0.705±0.004 | 0.490±0.016 | 0.512±0.014 | 0.540±0.014 | 0.362±0.015 | 0.352±0.010 | 0.381±0.009 | 0.309±0.012 | 0.285±0.014 | 0.285±0.011 |
| | UnionNET | 0.706±0.006 | 0.499±0.015 | 0.547±0.014 | 0.545±0.013 | 0.379±0.013 | 0.399±0.013 | 0.379±0.012 | 0.322±0.021 | 0.302±0.013 | 0.290±0.012 |
| | NRGNN | 0.710±0.006 | 0.498±0.015 | 0.546±0.015 | 0.538±0.011 | 0.382±0.016 | 0.412±0.016 | 0.377±0.012 | 0.336±0.021 | 0.309±0.018 | 0.284±0.009 |
| | RTGNN | 0.746±0.008 | 0.498±0.007 | 0.556±0.007 | 0.550±0.012 | 0.392±0.010 | 0.424±0.013 | 0.390±0.014 | 0.348±0.017 | 0.308±0.016 | 0.302±0.011 |
| | SUGRL | 0.730±0.005 | 0.493±0.011 | 0.541±0.011 | 0.544±0.010 | 0.376±0.009 | 0.421±0.009 | 0.388±0.009 | 0.339±0.010 | 0.305±0.010 | 0.300±0.009 |
| | MERIT | 0.740±0.007 | 0.496±0.012 | 0.536±0.012 | 0.542±0.010 | 0.383±0.011 | 0.425±0.011 | 0.387±0.008 | 0.344±0.014 | 0.301±0.014 | 0.295±0.009 |
| | SFA | 0.740±0.011 | 0.502±0.014 | 0.532±0.015 | 0.547±0.013 | 0.390±0.014 | 0.433±0.014 | 0.389±0.012 | 0.347±0.016 | 0.312±0.015 | 0.299±0.013 |
| | ARIEL | 0.727±0.007 | 0.500±0.008 | 0.550±0.013 | 0.548±0.008 | 0.391±0.009 | 0.427±0.012 | 0.389±0.014 | 0.349±0.014 | 0.307±0.013 | 0.299±0.013 |
| | Sel-Cl | 0.725±0.008 | 0.499±0.012 | 0.551±0.010 | 0.549±0.008 | 0.389±0.011 | 0.426±0.008 | 0.391±0.020 | 0.350±0.018 | 0.310±0.015 | 0.300±0.017 |
| | Jo-SRC | 0.730±0.005 | 0.500±0.013 | 0.555±0.011 | 0.551±0.011 | 0.394±0.013 | 0.425±0.013 | 0.393±0.013 | 0.351±0.013 | 0.305±0.018 | 0.303±0.013 |
| | GRAND+ | 0.756±0.004 | 0.497±0.010 | 0.553±0.010 | 0.552±0.011 | 0.390±0.013 | 0.422±0.013 | 0.387±0.013 | 0.348±0.013 | 0.309±0.014 | 0.302±0.012 |
| | GFSA | 0.743±0.006 | 0.495±0.012 | 0.546±0.012 | 0.546±0.011 | 0.386±0.011 | 0.418±0.011 | 0.386±0.012 | 0.342±0.013 | 0.308±0.015 | 0.298±0.012 |
| | HONGAT | 0.738±0.007 | 0.492±0.014 | 0.540±0.011 | 0.545±0.009 | 0.380±0.012 | 0.413±0.010 | 0.384±0.013 | 0.340±0.014 | 0.306±0.016 | 0.296±0.011 |
| | CRGNN | 0.751±0.006 | 0.497±0.011 | 0.552±0.010 | 0.549±0.012 | 0.389±0.014 | 0.423±0.013 | 0.388±0.012 | 0.347±0.015 | 0.310±0.014 | 0.301±0.012 |
| | CGNN | 0.741±0.007 | 0.493±0.013 | 0.544±0.012 | 0.546±0.010 | 0.385±0.013 | 0.419±0.012 | 0.385±0.011 | 0.343±0.013 | 0.307±0.013 | 0.297±0.012 |
| | LR-GCL | 0.757±0.010 | 0.520±0.013 | 0.581±0.013 | 0.570±0.007 | 0.410±0.014 | 0.455±0.014 | 0.406±0.012 | 0.369±0.012 | 0.335±0.014 | 0.318±0.010 |
| | LRA-LR-GCL | **0.762±0.010** | **0.533±0.013** | **0.597±0.013** | **0.588±0.007** | **0.430±0.014** | **0.472±0.014** | **0.423±0.012** | **0.392±0.012** | **0.352±0.014** | **0.335±0.010** |
| PubMed | GCN | 0.790±0.007 | 0.584±0.022 | 0.574±0.012 | 0.595±0.012 | 0.405±0.025 | 0.386±0.011 | 0.488±0.013 | 0.305±0.022 | 0.295±0.013 | 0.423±0.013 |
| | S²GC | 0.802±0.005 | 0.585±0.023 | 0.589±0.013 | 0.610±0.009 | 0.421±0.030 | 0.401±0.014 | 0.497±0.012 | 0.310±0.039 | 0.290±0.019 | 0.431±0.010 |
| | GCE | 0.792±0.009 | 0.589±0.018 | 0.581±0.011 | 0.590±0.014 | 0.430±0.012 | 0.399±0.012 | 0.491±0.010 | 0.311±0.021 | 0.301±0.011 | 0.424±0.012 |
| | UnionNET | 0.793±0.008 | 0.603±0.020 | 0.620±0.012 | 0.592±0.012 | 0.445±0.022 | 0.424±0.013 | 0.489±0.015 | 0.313±0.025 | 0.327±0.015 | 0.435±0.009 |
| | NRGNN | 0.797±0.008 | 0.602±0.022 | 0.618±0.013 | 0.603±0.008 | 0.443±0.012 | 0.434±0.012 | 0.499±0.009 | 0.330±0.023 | 0.325±0.013 | 0.433±0.011 |
| | RTGNN | 0.797±0.004 | 0.610±0.008 | 0.622±0.010 | 0.614±0.012 | 0.455±0.010 | 0.455±0.011 | 0.501±0.011 | 0.335±0.013 | 0.338±0.017 | 0.452±0.013 |
| | SUGRL | 0.819±0.005 | 0.603±0.013 | 0.615±0.013 | 0.615±0.010 | 0.445±0.011 | 0.441±0.011 | 0.501±0.007 | 0.321±0.009 | 0.321±0.009 | 0.446±0.010 |
| | MERIT | 0.801±0.004 | 0.593±0.011 | 0.612±0.011 | 0.613±0.011 | 0.447±0.012 | 0.443±0.012 | 0.497±0.009 | 0.328±0.011 | 0.323±0.011 | 0.445±0.009 |
| | ARIEL | 0.800±0.003 | 0.610±0.013 | 0.622±0.010 | 0.615±0.011 | 0.453±0.012 | 0.453±0.012 | 0.502±0.014 | 0.331±0.014 | 0.336±0.018 | 0.457±0.013 |
| | SFA | 0.804±0.010 | 0.596±0.011 | 0.615±0.011 | 0.609±0.011 | 0.447±0.014 | 0.446±0.017 | 0.499±0.014 | 0.330±0.011 | 0.327±0.011 | 0.447±0.014 |
| | Sel-Cl | 0.799±0.005 | 0.605±0.014 | 0.625±0.012 | 0.614±0.012 | 0.455±0.014 | 0.449±0.010 | 0.502±0.008 | 0.334±0.021 | 0.332±0.014 | 0.456±0.014 |
| | Jo-SRC | 0.801±0.005 | 0.613±0.010 | 0.624±0.013 | 0.617±0.013 | 0.453±0.008 | 0.455±0.013 | 0.504±0.013 | 0.330±0.015 | 0.334±0.018 | 0.459±0.018 |
| | GRAND+ | 0.845±0.006 | 0.610±0.011 | 0.624±0.013 | 0.617±0.013 | 0.453±0.008 | 0.453±0.011 | 0.503±0.010 | 0.331±0.014 | 0.337±0.013 | 0.458±0.014 |
| | GFSA | 0.823±0.005 | 0.608±0.012 | 0.621±0.011 | 0.616±0.009 | 0.450±0.013 | 0.452±0.012 | 0.500±0.010 | 0.333±0.013 | 0.334±0.011 | 0.455±0.012 |
| | HONGAT | 0.818±0.006 | 0.606±0.011 | 0.619±0.012 | 0.613±0.010 | 0.448±0.014 | 0.447±0.012 | 0.498±0.012 | 0.328±0.012 | 0.326±0.013 | 0.450±0.011 |
| | CRGNN | 0.829±0.005 | 0.612±0.010 | 0.623±0.009 | 0.618±0.011 | 0.452±0.011 | 0.455±0.013 | 0.503±0.009 | 0.335±0.013 | 0.333±0.014 | 0.457±0.012 |
| | CGNN | 0.822±0.006 | 0.607±0.013 | 0.620±0.011 | 0.615±0.010 | 0.449±0.012 | 0.451±0.014 | 0.499±0.010 | 0.332±0.014 | 0.330±0.012 | 0.454±0.013 |
| | LR-GCL | 0.845±0.009 | 0.637±0.014 | 0.645±0.015 | 0.637±0.011 | 0.479±0.011 | 0.484±0.013 | 0.526±0.011 | 0.356±0.011 | 0.360±0.012 | 0.482±0.014 |
| | LRA-LR-GCL | **0.846±0.009** | **0.652±0.014** | **0.662±0.015** | **0.655±0.011** | **0.498±0.011** | **0.503±0.013** | **0.544±0.011** | **0.379±0.011** | **0.379±0.012** | **0.498±0.014** |
| Coauthor-CS | GCN | 0.918±0.001 | 0.645±0.009 | 0.656±0.006 | 0.702±0.010 | 0.511±0.013 | 0.501±0.009 | 0.531±0.010 | 0.429±0.022 | 0.389±0.011 | 0.415±0.013 |
| | S²GC | 0.918±0.001 | 0.657±0.012 | 0.663±0.006 | 0.713±0.010 | 0.516±0.013 | 0.514±0.009 | 0.556±0.009 | 0.437±0.020 | 0.396±0.010 | 0.422±0.012 |
| | GCE | 0.922±0.003 | 0.662±0.017 | 0.659±0.007 | 0.705±0.014 | 0.515±0.016 | 0.502±0.007 | 0.539±0.009 | 0.443±0.017 | 0.389±0.012 | 0.412±0.011 |
| | UnionNET | 0.918±0.002 | 0.669±0.023 | 0.671±0.013 | 0.706±0.012 | 0.525±0.011 | 0.529±0.011 | 0.540±0.012 | 0.458±0.015 | 0.401±0.011 | 0.420±0.007 |
| | NRGNN | 0.919±0.002 | 0.678±0.014 | 0.689±0.009 | 0.705±0.012 | 0.545±0.021 | 0.556±0.011 | 0.546±0.011 | 0.461±0.012 | 0.410±0.012 | 0.417±0.007 |
| | RTGNN | 0.920±0.005 | 0.678±0.012 | 0.691±0.009 | 0.712±0.008 | 0.559±0.010 | 0.569±0.011 | 0.560±0.008 | 0.455±0.015 | 0.415±0.015 | 0.412±0.014 |
| | SUGRL | 0.922±0.005 | 0.675±0.010 | 0.695±0.010 | 0.714±0.006 | 0.550±0.011 | 0.560±0.011 | 0.561±0.007 | 0.449±0.011 | 0.411±0.011 | 0.429±0.008 |
| | MERIT | 0.924±0.004 | 0.679±0.011 | 0.689±0.008 | 0.709±0.005 | 0.552±0.014 | 0.562±0.014 | 0.562±0.011 | 0.452±0.013 | 0.403±0.013 | 0.426±0.005 |
| | ARIEL | 0.925±0.004 | 0.682±0.011 | 0.699±0.009 | 0.712±0.005 | 0.555±0.011 | 0.566±0.011 | 0.556±0.011 | 0.454±0.014 | 0.415±0.019 | 0.427±0.013 |
| | SFA | 0.925±0.009 | 0.682±0.011 | 0.690±0.012 | 0.715±0.012 | 0.555±0.015 | 0.567±0.014 | 0.565±0.013 | 0.458±0.013 | 0.402±0.013 | 0.429±0.015 |
| | Sel-Cl | 0.922±0.008 | 0.684±0.009 | 0.694±0.012 | 0.714±0.010 | 0.557±0.013 | 0.568±0.013 | 0.566±0.010 | 0.457±0.013 | 0.412±0.017 | 0.425±0.009 |
| | Jo-SRC | 0.921±0.005 | 0.684±0.011 | 0.695±0.004 | 0.709±0.007 | 0.560±0.011 | 0.566±0.011 | 0.561±0.009 | 0.456±0.013 | 0.410±0.018 | 0.428±0.010 |
| | GRAND+ | 0.927±0.004 | 0.682±0.011 | 0.693±0.006 | 0.715±0.008 | 0.554±0.008 | 0.568±0.013 | 0.557±0.011 | 0.455±0.012 | 0.416±0.013 | 0.428±0.011 |
| | GFSA | 0.923±0.004 | 0.679±0.010 | 0.687±0.009 | 0.711±0.009 | 0.550±0.012 | 0.559±0.011 | 0.558±0.010 | 0.453±0.014 | 0.410±0.012 | 0.426±0.011 |
| | HONGAT | 0.924±0.003 | 0.681±0.012 | 0.692±0.010 | 0.713±0.008 | 0.553±0.013 | 0.563±0.013 | 0.560±0.012 | 0.456±0.013 | 0.411±0.015 | 0.427±0.010 |
| | CRGNN | 0.926±0.005 | 0.683±0.011 | 0.690±0.011 | 0.712±0.007 | 0.551±0.015 | 0.561±0.012 | 0.559±0.011 | 0.454±0.012 | 0.412±0.014 | 0.426±0.012 |
| | CGNN | 0.925±0.006 | 0.680±0.012 | 0.689±0.012 | 0.710±0.010 | 0.549±0.014 | 0.560±0.012 | 0.557±0.012 | 0.452±0.013 | 0.409±0.015 | 0.425±0.012 |
| | LR-GCL | 0.933±0.006 | 0.699±0.015 | 0.721±0.011 | 0.742±0.015 | 0.575±0.014 | 0.595±0.018 | 0.588±0.015 | 0.469±0.015 | 0.438±0.015 | 0.453±0.017 |
| | LRA-LR-GCL | **0.934±0.006** | **0.714±0.015** | **0.736±0.011** | **0.758±0.015** | **0.594±0.014** | **0.612±0.018** | **0.606±0.015** | **0.489±0.015** | **0.453±0.015** | **0.470±0.017** |

SUGRL (Mo et al., 2022), and SFA (Zhang et al., 2023). We compare LR-GCL with attention-based GNNs, GFSA (Choi et al., 2024a) and HONGAT (Zhang et al., 2024a), which balance low-frequency information and high-frequency information learned from the graph. We also compare with CRGNN (Li et al., 2024) and CGNN (Yuan et al., 2023), which adopt graph contrastive learning to mitigate the label noise in the training data. To demonstrate the power of LR-GCL in learning robust node representation, we also compare LR-GCL with two robust contrastive learning baselines, Jo-SRC (Yao et al., 2021) and Sel-CL (Li et al., 2022),

which select clean samples for image data. Since their sample selection methods are general and not limited to the image domain, we adopt these two baselines to the graph domain in our experiments as detailed in Section C of the appendix.

**Experimental Results.** We first compare LR-GCL against competing methods for semi-supervised or transductive node classification on input with two types of label noise. To show the robustness of LR-GCL against label noise, we perform the experiments on graphs injected with different levels of label noise ranging from 40% to 80% with a step of 20%. We follow the widely used semi-supervised setting (Kipf & Welling, 2017) for node classification. In LR-GCL, we train a transductive classifier for node classification. Previous GCL methods, including MERIT, SUGRL, and SFA, train a linear layer for inductive classification on top of the node representations learned by contrastive learning without using test data in training. Because LR-GCL is a transductive classifier, for fair comparisons, we also train the compared GCL baselines with the same transductive classifier as that for LR-GCL and a two-layer GCN transductive classifier. The results with different types of classifiers are shown in Section 5.4. For all the baselines in our experiments that perform inductive classification when predicting the labels, we report their best results using their original inductive classifier and two types of transductive classifiers: the same transductive classifier as that for LR-GCL and a two-layer GCN transductive classifier.

Results on Cora, Citeseer, PubMed, and Coauthor-CS are shown in Table 3, where we report the means of the accuracy of 10 runs and the standard deviation. It is observed from the results that LR-GCL outperforms all the baselines. By selecting confident nodes and computing robust prototypes using BEC, LR-GCL outperforms all the baselines by an even larger margin with a larger label noise level. In addition, we compare LR-GCL with baselines for noisy input with attribute noise levels ranging from 40% to 80% with a step of 20%. The results for node classification with symmetric label noise, asymmetric label noise, and attribute noise on ogbn-arxiv, Wiki-CS, Amazon-Computers, and Amazon-Photos are shown in Table 8 in Section B.1, where we report the means of the accuracy of 10 runs and the standard deviation. It is observed that LR-GCL also outperforms all the baselines for node classification with both label noise and attribute noise on these four benchmark datasets.

## 5.4 Node Classification Results for GCL Methods with Different Types of Classifiers

Existing GCL methods, such as MERIT, SUGRL, and SFA, first train a graph encoder with graph contrastive learning objectives such as InfoNCE (Jin et al., 2021). After obtaining the node representation learned by contrastive learning, a linear layer for classification is trained in the supervised setting. In contrast, LR-GCL adopts a transductive classifier on top of the node representation obtained by contrastive learning. For fair comparisons with previous GCL methods, we also train the compared GCL baselines with the same transductive classifier as in LR-GCL and a two-layer transductive GCN classifier.

Table 4: Performance comparison for node classification by inductive linear classifier, transductive two-layer GCN classifier, and transductive classifier used in LR-GCL. The comparisons are performed on Cora.

| Methods | 0 | 40 | | | 60 | | | 80 | | |
|---|---|---|---|---|---|---|---|---|---|---|
| | - | Asymmetric | Symmetric | Attribute | Asymmetric | Symmetric | Attribute | Asymmetric | Symmetric | Attribute |
| SUGRL (original, inductive classifier) | 0.834±0.005 | 0.564±0.011 | 0.674±0.012 | 0.675±0.009 | 0.468±0.011 | 0.552±0.011 | 0.452±0.012 | 0.280±0.012 | 0.381±0.012 | 0.338±0.014 |
| SUGRL + transductive GCN | 0.833±0.006 | 0.562±0.013 | 0.675±0.015 | 0.673±0.012 | 0.470±0.011 | 0.551±0.011 | 0.454±0.012 | 0.280±0.012 | 0.380±0.012 | 0.340±0.014 |
| SUGRL + linear transductive classifier | 0.836±0.007 | 0.568±0.013 | 0.677±0.010 | 0.674±0.011 | 0.472±0.011 | 0.555±0.011 | 0.457±0.012 | 0.284±0.012 | 0.383±0.012 | 0.341±0.014 |
| MERIT (original, inductive classifier) | 0.831±0.005 | 0.560±0.008 | 0.670±0.008 | 0.671±0.009 | 0.467±0.013 | 0.547±0.013 | 0.450±0.014 | 0.277±0.013 | 0.385±0.013 | 0.335±0.009 |
| MERIT + transductive GCN | 0.831±0.007 | 0.562±0.011 | 0.668±0.013 | 0.672±0.014 | 0.466±0.013 | 0.549±0.015 | 0.451±0.016 | 0.276±0.012 | 0.382±0.014 | 0.337±0.013 |
| MERIT + linear transductive classifier | 0.833±0.003 | 0.562±0.014 | 0.673±0.012 | 0.673±0.011 | 0.466±0.015 | 0.546±0.016 | 0.453±0.017 | 0.280±0.016 | 0.386±0.011 | 0.336±0.014 |
| SFA (original, inductive classifier) | 0.839±0.010 | 0.564±0.011 | 0.677±0.013 | 0.676±0.015 | 0.473±0.014 | 0.549±0.014 | 0.457±0.014 | 0.282±0.016 | 0.389±0.013 | 0.344±0.017 |
| SFA + transductive GCN | 0.837±0.013 | 0.565±0.011 | 0.673±0.017 | 0.673±0.018 | 0.474±0.016 | 0.551±0.015 | 0.453±0.018 | 0.277±0.016 | 0.389±0.015 | 0.343±0.019 |
| SFA + linear transductive classifier | 0.841±0.015 | 0.566±0.013 | 0.678±0.014 | 0.679±0.014 | 0.477±0.015 | 0.552±0.012 | 0.456±0.016 | 0.284±0.017 | 0.391±0.015 | 0.348±0.019 |
| LR-GCL | 0.757±0.010 | 0.520±0.013 | 0.581±0.013 | 0.570±0.007 | 0.410±0.014 | 0.455±0.014 | 0.406±0.012 | 0.369±0.012 | 0.335±0.014 | 0.318±0.010 |
| LRA-LR-GCL | **0.762±0.010** | **0.533±0.013** | **0.597±0.013** | **0.588±0.007** | **0.430±0.014** | **0.472±0.014** | **0.423±0.012** | **0.392±0.012** | **0.352±0.014** | **0.335±0.010** |

## 5.5 Study in the Kernel Complexity and the Upper Bound of the Test Loss

In this section, we compute the kernel complexity (KC) for the gram matrix of node representations learned by LR-GCL and the competing GCL methods on different datasets with asymmetric label noise of level 40 by Equation (5) in Theorem 4.1. The results are shown in Table 5. It is observed that the gram matrix of the node representations learned by LR-GCL exhibits much lower complexity, which suggests that the transduc-

tive classifiers trained on the node representations learned by LR-GCL have lower generalization errors on the unlabeled nodes. Furthermore, we compare each term in the upper bound of the test loss in Equation 4, including $L_1(\mathbf{K}, \tilde{\mathbf{Y}}, t)$, $L_2(\mathbf{K}, \mathbf{N}, t)$, and KC($\mathbf{K}$), for the gram matrix of the node representation learned by different methods in Table 6. It is observed that LR-GCL and LRA-LR-GCL exhibit a significantly lower value in each of the terms than the competing baseline methods, demonstrating the better generalization capability of LR-GCL and LRA-LR-GCL for semi-supervised node classification even under the presence of label noise.

Table 5: Comparisons in complexity of kernels. The evaluation is performed on semi-supervised node classification with 40% of symmetric label noise.

| Datasets | | MERIT | SFA | Jo-SRC | GCN | GFSA | HONGAT | LR-GCL | LRA-LR-GCL |
|---|---|---|---|---|---|---|---|---|---|
| Cora | KC | 0.37 | 0.42 | 0.48 | 0.44 | 0.35 | 0.40 | 0.20 | 0.18 |
| | $r_0$ | 1420 | 1478 | 1665 | 1511 | 1262 | 1450 | 440 | 395 |
| Citeseer | KC | 0.47 | 0.45 | 0.55 | 0.64 | 0.47 | 0.50 | 0.24 | 0.21 |
| | $r_0$ | 1214 | 1180 | 1405 | 1590 | 1224 | 1285 | 405 | 369 |
| PubMed | KC | 0.54 | 0.50 | 0.62 | 0.71 | 0.52 | 0.66 | 0.30 | 0.28 |
| | $r_0$ | 1644 | 1562 | 1785 | 1993 | 1588 | 1874 | 1197 | 1090 |
| Wiki-CS | KC | 0.42 | 0.44 | 0.40 | 0.49 | 0.43 | 0.45 | 0.19 | 0.17 |
| | $r_0$ | 1805 | 1993 | 1746 | 2130 | 1842 | 2048 | 970 | 904 |
| Amazon-Computers | KC | 0.39 | 0.37 | 0.40 | 0.45 | 0.35 | 0.37 | 0.12 | 0.11 |
| | $r_0$ | 1450 | 1428 | 1489 | 1632 | 1370 | 1415 | 874 | 820 |
| Amazon-Photos | KC | 0.38 | 0.38 | 0.43 | 0.47 | 0.39 | 0.41 | 0.14 | 0.12 |
| | $r_0$ | 1872 | 1884 | 1990 | 2145 | 1895 | 1921 | 750 | 722 |
| Coauthor-CS | KC | 0.29 | 0.28 | 0.32 | 0.34 | 0.31 | 0.32 | 0.12 | 0.11 |
| | $r_0$ | 1774 | 1725 | 1896 | 1903 | 1872 | 1890 | 1120 | 1039 |
| ogbn-arxiv | KC | 0.12 | 0.13 | 0.12 | 0.14 | 0.12 | 0.13 | 0.05 | 0.05 |
| | $r_0$ | 1860 | 1936 | 1852 | 1996 | 1845 | 1920 | 1354 | 1328 |

Table 6: Comparisons on $L_1(\mathbf{K}, \tilde{\mathbf{Y}}, t)$, $L_2(\mathbf{K}, \mathbf{N}, t)$, KC($\mathbf{K}$) and the value of the upper bound of the test loss from Theorem 4.1. The evaluation is performed on semi-supervised node classification with 40% of symmetric label noise. The lowest values for each dataset in the table are bold, and the second-lowest values are underlined.

| Datasets | | MERIT | SFA | Jo-SRC | GCN | GFSA | HONGAT | LR-GCL | LRA-LR-GCL |
|---|---|---|---|---|---|---|---|---|---|
| Cora | $L_1$ | 5.24 | 6.04 | 6.50 | 7.38 | 6.44 | 6.38 | 3.72 | **3.65** |
| | $L_2$ | 4.92 | 4.95 | 5.05 | 5.24 | 3.80 | 4.25 | 2.97 | **2.72** |
| | KC | 0.37 | 0.42 | 0.48 | 0.44 | 0.35 | 0.40 | 0.20 | **0.18** |
| | Upper Bound | 10.68 | 11.59 | 12.18 | 13.22 | 10.80 | 11.25 | 7.05 | **6.74** |
| Citeseer | $L_1$ | 4.72 | 4.85 | 4.92 | 5.10 | 4.54 | 4.69 | 4.02 | **3.95** |
| | $L_2$ | 4.33 | 4.69 | 4.42 | 5.08 | 4.20 | 4.42 | 3.75 | **3.60** |
| | KC | 0.47 | 0.45 | 0.55 | 0.64 | 0.47 | 0.50 | 0.24 | **0.21** |
| | Upper Bound | 9.77 | 10.21 | 10.17 | 11.07 | 9.40 | 9.84 | 8.20 | **7.97** |
| PubMed | $L_1$ | 3.97 | 4.02 | 4.11 | 4.35 | 4.26 | 3.95 | 3.38 | **3.40** |
| | $L_2$ | 2.69 | 2.54 | 2.60 | 2.88 | 2.98 | 2.85 | 2.32 | **2.26** |
| | KC | 0.54 | 0.50 | 0.62 | 0.71 | 0.52 | 0.66 | 0.30 | **0.28** |
| | Upper Bound | 7.44 | 7.28 | 7.59 | 8.15 | 7.99 | 7.63 | 6.25 | **6.16** |

## 5.6 Ablation Study on the Rank in the Truncated Nuclear Norm

We perform ablation study on the value of rank $r_0$ in the TNN $\|\mathbf{K}\|_{r_0}$ in the loss function (2) of LR-GCL. It is observed from Table 7 that the performance of our LR-GCL is consistently close to the best performance among all the choices of the rank when $r_0$ is between $0.1 \min\{N, d\}$ and $0.3 \min\{N, d\}$.

Furthermore, we compare the training time of LR-GCL with competing baselines in Table 9 in Section B.2 of the appendix. We study the effectiveness of LR-GCL and LRA-LR-GCL on the heterophilic graphs in Section B.4 of the appendix. The node classification results in Table 10 show that both LR-GCL and LRA-LR-GCL remain effective on heterophilic graphs in combating the label noise and the attribute noise for node classification.

## 5.7 Visualization of the Low Frequency Property (LFP) by Eigen-Projections

The eigen-projection and energy concentration on Cora, Citeseer, and Pubmed are illustrated in Figure 2. The eigen-projection and energy concentration on Coauthor-CS, Amazon Computers, Amazon Photos, and ogbn-arxiv are illustrated in Figure 3 in Section B.3 of the supplementary. More eigen-projection and energy

Table 7: Ablation study on the value of rank $r_0$ in the optimization problem (3) on Cora with different levels of asymmetric and symmetric label noise. The accuracy with the optimal rank is shown in the last row. The accuracy difference against the optimal rank is shown for other ranks.

| Rank | Noise Type | | | | | | |
|---|---|---|---|---|---|---|---|
| | 0 | 40 | | 60 | | 80 | |
| | - | Asymmetric | Symmetric | Asymmetric | Symmetric | Asymmetric | Symmetric |
| $0.1 \min\{N, d\}$ | -0.002 | -0.001 | -0.002 | -0.002 | -0.001 | -0.001 | -0.000 |
| $0.2 \min\{N, d\}$ | -0.000 | -0.000 | -0.000 | -0.000 | -0.000 | -0.000 | -0.000 |
| $0.3 \min\{N, d\}$ | -0.000 | -0.000 | -0.001 | -0.002 | -0.001 | -0.000 | -0.001 |
| $0.4 \min\{N, d\}$ | -0.001 | -0.003 | -0.002 | -0.001 | -0.002 | -0.002 | -0.002 |
| $0.5 \min\{N, d\}$ | -0.001 | -0.002 | -0.003 | -0.003 | -0.003 | -0.001 | -0.002 |
| $0.6 \min\{N, d\}$ | -0.003 | -0.002 | -0.002 | -0.003 | -0.002 | -0.002 | -0.003 |
| $0.7 \min\{N, d\}$ | -0.003 | -0.004 | -0.003 | -0.004 | -0.004 | -0.004 | -0.005 |
| $0.8 \min\{N, d\}$ | -0.002 | -0.005 | -0.006 | -0.006 | -0.006 | -0.007 | -0.007 |
| $0.9 \min\{N, d\}$ | -0.004 | -0.004 | -0.005 | -0.007 | -0.008 | -0.008 | -0.006 |
| $\min\{N, d\}$ | -0.004 | -0.004 | -0.007 | -0.007 | -0.008 | -0.010 | -0.008 |
| optimal | 0.858 | 0.589 | 0.713 | 0.492 | 0.587 | 0.306 | 0.419 |

(a) Cora   (b) Citeseer   (c) Pubmed

Figure 2: Eigen-projection (first row) and signal concentration ratio (second row) on Cora, Citeseer, and Pubmed. To compute the eigen-projection, we first calculate the eigenvectors $\mathbf{U}$ of the feature gram matrix $\mathbf{K} = \mathbf{HH}^\top$, then the eigen-projection value is computed by $p_r = \frac{1}{C} \sum_{c=1}^{C} \left\| \mathbf{U}^{(r)\top} \tilde{\mathbf{Y}}^{(c)} \right\|_2^2 / \left\| \tilde{\mathbf{Y}}^{(c)} \right\|_2^2$ for $r \in [N]$, where $C$ is the number of classes, and $\tilde{\mathbf{Y}} \in \{0, 1\}^{N \times C}$ is the one-hot clean labels of all the nodes, $\tilde{\mathbf{Y}}^{(c)}$ is the $c$-th column of $\tilde{\mathbf{Y}}$. We let $\mathbf{p} = [p_1, \ldots, p_N] \in \mathbb{R}^N$. With the presence of label noise $\mathbf{N} \in \mathbb{R}^{N \times C}$, the observed label matrix is $\mathbf{Y} = \tilde{\mathbf{Y}} + \mathbf{N}$. The eigen-projection $p_r$ reflects the amount of the signal projected onto the $r$-th eigenvector of $\mathbf{K}$, and the signal concentration ratio of a rank $r$ reflects the proportion of signal projected onto the top $r$ eigenvectors of $\mathbf{K}$. The signal concentration ratio for rank $r$ is computed by $\left\| \mathbf{p}^{(1:r)} \right\|_1$, where $\mathbf{p}^{(1:r)}$ contains the first $r$ elements of $\mathbf{p}$. It is observed from the red curves in the first row that the projection of the ground truth clean labels mostly concentrates on the top eigenvectors of $\mathbf{K}$. On the other hand, the projection of label noise, computed by $\frac{1}{C} \sum_{c=1}^{C} \left\| \mathbf{U}^\top \mathbf{N}^{(c)} \right\|_2^2 / \left\| \mathbf{Y}^{(c)} \right\|_2^2 \in \mathbb{R}^N$, is relatively uniform over all the eigenvectors, as illustrated by the blue curves in the first row. The study in this figure is performed for asymmetric label noise with a noise level of 60%. By the rank $r = 0.2 \min\{N, d\}$, the signal concentration ratio of $\tilde{\mathbf{Y}}$ for Cora, Citeseer, and Pubmed are 0.844, 0.809, and 0.784 respectively. We refer to such property as the **low frequency property**, which suggests that we can learn a low-rank portion of the observed label $\mathbf{Y}$ which covers most information in the ground truth clean label while only learning a small portion of the label noise. Figure 3 in the appendix further illustrates the low frequency property on more datasets.

concentration on the heterophilic graphs illustrated in Figure 4 in Section B.4 of the appendix demonstrate that LFP also exists in the heterophilic graphs.

## 6 Conclusions

In this paper, we propose a novel GCL encoder termed Low-Rank Graph Contrastive Learning (LR-GCL). LR-GCL is a robust GCL encoder which produces low-rank features inspired by the low frequency property of universal graph datasets and the sharp generalization bound for transductive learning. LR-GCL is trained with prototypical GCL with the TNN as the regularization term. We evaluate the performance of LR-GCL with comparison to competing baselines on semi-supervised or transductive node classification, where graph data are corrupted with noise in either the labels for the node attributes. Extensive experimental results demonstrate that LR-GCL generates more robust node representations with better performance than the current state-of-the-art node representation learning methods.

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

# A    Theoretical Results

We present the proof of Theorem 4.1 in this section.

**Proof of Theorem 4.1**. Define $\mathbf{N} := \mathbf{Y} - \tilde{\mathbf{Y}} \in \mathbb{R}^N$ as the label noise. It can be verified that at the $t$-th iteration of gradient descent for $t \geq 1$, we have

$$
\begin{aligned}
\mathbf{W}^{(t)} &= \mathbf{W}^{(t-1)} - \eta \left[\mathbf{H}\right]_{\mathcal{L}}^{\top} \left[\mathbf{H}\mathbf{W}^{(t-1)} - \mathbf{Y}\right]_{\mathcal{L}} \\
&= \mathbf{W}^{(t-1)} - \eta \left[\mathbf{H}\right]_{\mathcal{L}}^{\top} \left[\mathbf{H}\mathbf{W}^{(t-1)} - \tilde{\mathbf{Y}}\right]_{\mathcal{L}} + \eta \left[\mathbf{H}\right]_{\mathcal{L}}^{\top} \left[\mathbf{N}\right]_{\mathcal{L}}.
\end{aligned}
\tag{7}
$$

It follows by (7) that

$$
\left[\mathbf{H}\right]_{\mathcal{L}} \mathbf{W}^{(t)} = \left[\mathbf{H}\right]_{\mathcal{L}} \mathbf{W}^{(t-1)} - \eta \mathbf{K}_{\mathcal{L},\mathcal{L}} \left[\mathbf{H}\mathbf{W}^{(t-1)} - \tilde{\mathbf{Y}}\right]_{\mathcal{L}} + \eta \left[\mathbf{K}\right]_{\mathcal{L},\mathcal{L}} \left[\mathbf{N}\right]_{\mathcal{L}},
\tag{8}
$$

where $\mathbf{K}_{\mathcal{L},\mathcal{L}} := \left[\mathbf{H}\right]_{\mathcal{L}} \left[\mathbf{H}\right]_{\mathcal{L}}^{\top} \in \mathbb{R}^{m \times m}$. With $\mathbf{F}(\mathbf{W}, t) = \mathbf{H}\mathbf{W}^{(t)}$, it follows by (8) that

$$
\left[\mathbf{F}(\mathbf{W}, t) - \tilde{\mathbf{Y}}\right]_{\mathcal{L}} = \left(\mathbf{I}_m - \eta \left[\mathbf{K}\right]_{\mathcal{L},\mathcal{L}}\right) \left[\mathbf{F}(\mathbf{W}, t-1) - \tilde{\mathbf{Y}}\right]_{\mathcal{L}} + \eta \left[\mathbf{K}\right]_{\mathcal{L},\mathcal{L}} \left[\mathbf{N}\right]_{\mathcal{L}}.
$$

It follows from the above equality and the recursion that

$$
\left[\mathbf{F}(\mathbf{W}, t) - \tilde{\mathbf{Y}}\right]_{\mathcal{L}} = - \left(\mathbf{I}_m - \eta \left[\mathbf{K}\right]_{\mathcal{L},\mathcal{L}}\right)^t \left[\tilde{\mathbf{Y}}\right]_{\mathcal{L}} + \eta \left[\mathbf{K}\right]_{\mathcal{L},\mathcal{L}} \sum_{t'=0}^{t-1} \left(\mathbf{I}_m - \eta \left[\mathbf{K}\right]_{\mathcal{L},\mathcal{L}}\right)^{t'} \left[\mathbf{N}\right]_{\mathcal{L}}
\tag{9}
$$

We apply (Yang, 2023, Corollary 3.7) to obtain the following bound for the test loss $\frac{1}{u}\left\|\left[\mathbf{F}(\mathbf{W}, t) - \tilde{\mathbf{Y}}\right]_{\mathcal{U}}\right\|_{\mathrm{F}}^2$:

$$
\frac{1}{u}\left\|\left[\mathbf{F}(\mathbf{W}, t) - \tilde{\mathbf{Y}}\right]_{\mathcal{U}}\right\|_{\mathrm{F}}^2 \leq \frac{c_0}{m}\left\|\left[\mathbf{F}(\mathbf{W}, t) - \tilde{\mathbf{Y}}\right]_{\mathcal{L}}\right\|_{\mathrm{F}}^2 + c_0 \min_{0 \leq Q \leq n} r(u, m, Q) + \frac{c_0 x}{u},
\tag{10}
$$

with

$$
r(u, m, Q) := Q\left(\frac{1}{u} + \frac{1}{m}\right) + \left(\sqrt{\frac{\sum_{q=Q+1}^{N} \widehat{\lambda}_q}{u}} + \sqrt{\frac{\sum_{q=Q+1}^{N} \widehat{\lambda}_q}{m}}\right),
$$

where $c_0$ is a positive constant depending on $\mathbf{U}$, $\left\{\widehat{\lambda}_i\right\}_{i=1}^r$, and $\tau_0$ with $\tau_0^2 = \max_{i \in [N]} \mathbf{K}_{ii}$.

It follows from (9) and (10) that for every $r_0 \in [0, n]$, we have

$$\frac{1}{u} \left\| \left[ \mathbf{F}(\mathbf{W}, t) - \tilde{\mathbf{Y}} \right]_{\mathcal{U}} \right\|_{\mathrm{F}}^2$$

$$\leq \frac{c_0}{m} \left\| \left( \mathbf{I}_m - \eta \left[ \mathbf{K} \right]_{\mathcal{L}, \mathcal{L}} \right)^t \left[ \tilde{\mathbf{Y}} \right]_{\mathcal{L}} \right\|_{\mathrm{F}}^2 + c_0 r_0 \left( \frac{1}{u} + \frac{1}{m} \right) + c_0 \left( \sqrt{\frac{\sum_{q=r_0+1}^{N} \widehat{\lambda}_q}{u}} + \sqrt{\frac{\sum_{q=r_0+1}^{N} \widehat{\lambda}_q}{m}} \right) + \frac{c_0 x}{u}$$

$$\overset{\text{①}}{\leq} \frac{2c_0}{m} \left\| \left( \mathbf{I}_m - \eta \left[ \mathbf{K} \right]_{\mathcal{L}, \mathcal{L}} \right)^t \left[ \tilde{\mathbf{Y}} \right]_{\mathcal{L}} \right\|_{\mathrm{F}}^2 + \frac{2c_0}{m} \left\| \eta \left[ \mathbf{K} \right]_{\mathcal{L}, \mathcal{L}} \sum_{t'=0}^{t-1} \left( \mathbf{I}_m - \eta \left[ \mathbf{K} \right]_{\mathcal{L}, \mathcal{L}} \right)^{t'} \left[ \mathbf{N} \right]_{\mathcal{L}} \right\|_{\mathrm{F}}^2$$

$$+ c_0 r_0 \left( \frac{1}{u} + \frac{1}{m} \right) + c_0 \sqrt{\|\mathbf{K}\|_{r_0}} \left( \sqrt{\frac{1}{u}} + \sqrt{\frac{1}{m}} \right) + \frac{c_0 x}{u}, \tag{11}$$

where ① follows from the Cauchy-Schwarz inequality, (9), and $\sum_{q=r_0+1}^{N} \widehat{\lambda}_q = \|\mathbf{K}\|_{r_0}$. (4) then follows directly from (11). $\qquad\square$

## B  Additional Experiment Results

### B.1  Additional Node Classification Results

The results for node classification with symmetric label noise, asymmetric label noise, and attribute noise on ogbn-arxiv, Wiki-CS, Amazon-Computers, and Amazon-Photos are shown in Table 8 in Section B.1, where we report the means of the accuracy of 10 runs and the standard deviation. It is observed that LR-GCL also outperforms all the baselines for node classification with both label noise and attribute noise on these four benchmark datasets. For example, LRA-GCL outperforms the best baseline method by 2.3% in node classification accuracy on PubMed with 80% symmetric label noise.

### B.2  Training Time Comparison

In this section, we compare the training time of LR-GCL against other baseline methods on all benchmark datasets. The training time of LR-GCL includes the training time of robust graph contrastive learning, the time of the SVD computation of the kernel, and the training time of the transductive classifier. For the competing GCL methods, we include both the training time of the GCL encoder and the downstream classifier. The training time is evaluated on one 80 GB A100 GPU. The results are shown in Table 9. It is observed that the LR-GCL takes a similar training time as the competing GCL methods, such as SFA and MERIT.

### B.3  Eigen-Projection and Concentration Entropy Analysis on Additional Datasets

Figure 3 illustrates the eigen-projection and signal concentration ratio for Coauthor-CS, Amazon-Computers, Amazon-Photos, and ogbn-arxiv.

### B.4  Evaluation on Heterophilic Graphs

In this section, we study the effectiveness of LR-GCL for semi-supervised node classification on two widely used heterophilic graph datasets, namely Texas and Chameleon (Pei et al., 2020). We first study the LFP on Texas and Chameleon by the eigen-projection and signal concentration ratio illustrated in Figure 4. It is observed that LFP also exists in the heterophilic graph datasets similar to that in the homophily datasets. The study in this figure is performed for asymmetric label noise with a noise level of 60%. By the rank of $0.2 \min \{N, d\}$, the concentration entropy on Chameleon and Texas are 0.762 and 0.725. Next, we perform the semi-supervised node classification experiments on Texas and Chameleon following the setting

Table 8: Performance comparison for node classification on Texas and Chameleon with asymmetric label noise, symmetric label noise, and attribute noise.

| Dataset | Methods | Noise Type | | | | | | | | | |
| | | 0 | 40 | | | 60 | | | 80 | | |
| | | - | Asymmetric | Symmetric | Attribute | Asymmetric | Symmetric | Attribute | Asymmetric | Symmetric | Attribute |
|---|---|---|---|---|---|---|---|---|---|---|---|
| ogbn-arxiv | GCN | 0.717±0.003 | 0.401±0.014 | 0.421±0.014 | 0.478±0.010 | 0.336±0.011 | 0.346±0.021 | 0.339±0.012 | 0.286±0.022 | 0.256±0.010 | 0.294±0.013 |
| | S²GC | 0.712±0.003 | 0.417±0.017 | 0.429±0.014 | 0.492±0.010 | 0.344±0.016 | 0.353±0.031 | 0.343±0.009 | 0.297±0.023 | 0.266±0.013 | 0.284±0.012 |
| | GCE | 0.720±0.004 | 0.410±0.018 | 0.428±0.008 | 0.480±0.014 | 0.348±0.019 | 0.344±0.019 | 0.342±0.015 | 0.310±0.014 | 0.260±0.011 | 0.275±0.015 |
| | UnionNET | 0.724±0.006 | 0.429±0.021 | 0.449±0.007 | 0.485±0.012 | 0.362±0.018 | 0.367±0.008 | 0.340±0.009 | 0.332±0.019 | 0.269±0.013 | 0.280±0.012 |
| | NRGNN | 0.721±0.006 | 0.449±0.014 | 0.466±0.009 | 0.485±0.012 | 0.371±0.020 | 0.379±0.008 | 0.342±0.011 | 0.330±0.018 | 0.271±0.018 | 0.300±0.010 |
| | RTGNN | 0.718±0.004 | 0.443±0.012 | 0.464±0.012 | 0.484±0.014 | 0.380±0.011 | 0.384±0.013 | 0.340±0.017 | 0.335±0.011 | 0.285±0.015 | 0.301±0.006 |
| | SUGRL | 0.693±0.002 | 0.439±0.010 | 0.467±0.010 | 0.480±0.012 | 0.365±0.013 | 0.385±0.011 | 0.341±0.009 | 0.327±0.011 | 0.275±0.011 | 0.295±0.011 |
| | MERIT | 0.717±0.004 | 0.442±0.009 | 0.463±0.009 | 0.483±0.010 | 0.368±0.011 | 0.381±0.011 | 0.341±0.012 | 0.324±0.012 | 0.272±0.010 | 0.304±0.009 |
| | ARIEL | 0.717±0.004 | 0.448±0.013 | 0.471±0.013 | 0.482±0.011 | 0.379±0.014 | 0.384±0.015 | 0.342±0.015 | 0.334±0.014 | 0.280±0.013 | 0.300±0.010 |
| | SFA | 0.718±0.009 | 0.445±0.012 | 0.463±0.013 | 0.486±0.012 | 0.368±0.014 | 0.378±0.014 | 0.338±0.015 | 0.325±0.014 | 0.273±0.012 | 0.302±0.013 |
| | Sel-Cl | 0.719±0.002 | 0.447±0.007 | 0.469±0.007 | 0.486±0.010 | 0.375±0.008 | 0.389±0.025 | 0.344±0.013 | 0.331±0.008 | 0.284±0.019 | 0.304±0.012 |
| | Jo-SRC | 0.715±0.005 | 0.445±0.011 | 0.466±0.009 | 0.481±0.010 | 0.377±0.013 | 0.387±0.013 | 0.340±0.013 | 0.333±0.013 | 0.282±0.018 | 0.297±0.009 |
| | GRAND+ | 0.725±0.004 | 0.445±0.008 | 0.466±0.011 | 0.481±0.011 | 0.378±0.010 | 0.385±0.012 | 0.344±0.010 | 0.332±0.010 | 0.282±0.016 | 0.303±0.009 |
| | GFSA | 0.719±0.004 | 0.443±0.012 | 0.460±0.010 | 0.482±0.011 | 0.370±0.012 | 0.379±0.012 | 0.342±0.011 | 0.328±0.012 | 0.278±0.013 | 0.299±0.011 |
| | HONGAT | 0.716±0.005 | 0.440±0.011 | 0.458±0.012 | 0.480±0.012 | 0.366±0.013 | 0.373±0.013 | 0.339±0.012 | 0.324±0.014 | 0.276±0.014 | 0.296±0.012 |
| | CRGNN | 0.721±0.003 | 0.446±0.010 | 0.465±0.010 | 0.483±0.009 | 0.372±0.010 | 0.382±0.011 | 0.343±0.010 | 0.330±0.012 | 0.281±0.012 | 0.302±0.010 |
| | CGNN | 0.717±0.006 | 0.441±0.013 | 0.462±0.011 | 0.481±0.010 | 0.368±0.014 | 0.376±0.012 | 0.340±0.011 | 0.326±0.015 | 0.277±0.013 | 0.298±0.012 |
| | LR-GCL | 0.728±0.006 | 0.472±0.013 | 0.492±0.011 | 0.508±0.014 | 0.405±0.014 | 0.411±0.012 | 0.405±0.012 | 0.359±0.015 | 0.307±0.013 | 0.335±0.013 |
| | LRA-LR-GCL | 0.731±0.006 | 0.487±0.013 | 0.507±0.011 | 0.523±0.014 | 0.423±0.014 | 0.430±0.012 | 0.423±0.012 | 0.374±0.015 | 0.332±0.013 | 0.350±0.013 |
| Wiki-CS | GCN | 0.918±0.001 | 0.645±0.009 | 0.656±0.006 | 0.702±0.010 | 0.511±0.013 | 0.501±0.009 | 0.531±0.010 | 0.429±0.022 | 0.389±0.011 | 0.415±0.013 |
| | S²GC | 0.918±0.001 | 0.657±0.012 | 0.663±0.006 | 0.713±0.010 | 0.516±0.013 | 0.514±0.009 | 0.556±0.009 | 0.437±0.020 | 0.396±0.010 | 0.422±0.012 |
| | GCE | 0.922±0.003 | 0.662±0.017 | 0.659±0.007 | 0.705±0.014 | 0.515±0.016 | 0.502±0.007 | 0.539±0.009 | 0.443±0.017 | 0.389±0.012 | 0.412±0.011 |
| | UnionNET | 0.918±0.002 | 0.669±0.023 | 0.671±0.013 | 0.706±0.012 | 0.525±0.011 | 0.529±0.011 | 0.540±0.012 | 0.458±0.015 | 0.401±0.011 | 0.420±0.007 |
| | NRGNN | 0.919±0.002 | 0.678±0.014 | 0.689±0.009 | 0.705±0.012 | 0.545±0.021 | 0.556±0.011 | 0.546±0.011 | 0.461±0.012 | 0.410±0.012 | 0.417±0.007 |
| | RTGNN | 0.920±0.005 | 0.678±0.012 | 0.691±0.009 | 0.712±0.008 | 0.559±0.010 | 0.569±0.011 | 0.560±0.008 | 0.455±0.015 | 0.415±0.015 | 0.412±0.014 |
| | SUGRL | 0.922±0.005 | 0.675±0.010 | 0.695±0.010 | 0.714±0.006 | 0.550±0.011 | 0.560±0.011 | 0.561±0.007 | 0.449±0.011 | 0.411±0.011 | 0.429±0.008 |
| | MERIT | 0.924±0.004 | 0.679±0.011 | 0.689±0.008 | 0.709±0.005 | 0.552±0.014 | 0.562±0.014 | 0.562±0.011 | 0.452±0.013 | 0.403±0.013 | 0.426±0.005 |
| | ARIEL | 0.925±0.004 | 0.682±0.011 | 0.699±0.009 | 0.712±0.005 | 0.555±0.011 | 0.566±0.011 | 0.556±0.011 | 0.454±0.014 | 0.415±0.019 | 0.427±0.013 |
| | SFA | 0.925±0.009 | 0.682±0.011 | 0.690±0.012 | 0.715±0.012 | 0.555±0.015 | 0.567±0.014 | 0.565±0.013 | 0.458±0.013 | 0.402±0.013 | 0.429±0.015 |
| | Sel-Cl | 0.922±0.008 | 0.684±0.009 | 0.694±0.012 | 0.714±0.010 | 0.557±0.013 | 0.568±0.013 | 0.566±0.010 | 0.457±0.013 | 0.412±0.017 | 0.425±0.009 |
| | Jo-SRC | 0.921±0.005 | 0.684±0.011 | 0.695±0.004 | 0.709±0.007 | 0.560±0.011 | 0.566±0.011 | 0.561±0.009 | 0.456±0.013 | 0.410±0.018 | 0.428±0.010 |
| | GRAND+ | 0.927±0.004 | 0.682±0.011 | 0.693±0.006 | 0.715±0.008 | 0.554±0.008 | 0.568±0.013 | 0.557±0.011 | 0.455±0.012 | 0.416±0.013 | 0.428±0.011 |
| | GFSA | 0.923±0.004 | 0.680±0.012 | 0.691±0.008 | 0.711±0.010 | 0.553±0.010 | 0.562±0.011 | 0.560±0.010 | 0.453±0.014 | 0.408±0.012 | 0.423±0.010 |
| | HONGAT | 0.921±0.003 | 0.674±0.014 | 0.685±0.010 | 0.707±0.011 | 0.546±0.012 | 0.553±0.010 | 0.552±0.010 | 0.448±0.014 | 0.404±0.013 | 0.419±0.012 |
| | CRGNN | 0.924±0.005 | 0.683±0.011 | 0.696±0.008 | 0.713±0.008 | 0.557±0.010 | 0.565±0.012 | 0.564±0.009 | 0.456±0.013 | 0.411±0.012 | 0.426±0.010 |
| | CGNN | 0.920±0.004 | 0.677±0.010 | 0.688±0.009 | 0.710±0.011 | 0.549±0.011 | 0.559±0.013 | 0.558±0.010 | 0.451±0.015 | 0.406±0.012 | 0.421±0.009 |
| | LR-GCL | 0.933±0.006 | 0.699±0.015 | 0.721±0.011 | 0.742±0.015 | 0.575±0.014 | 0.595±0.018 | 0.588±0.015 | 0.469±0.015 | 0.438±0.015 | 0.453±0.017 |
| | LRA-LR-GCL | 0.936±0.006 | 0.714±0.015 | 0.736±0.011 | 0.758±0.015 | 0.594±0.014 | 0.612±0.018 | 0.606±0.015 | 0.489±0.015 | 0.453±0.015 | 0.470±0.017 |
| Amazon-Computers | GCN | 0.815±0.005 | 0.547±0.015 | 0.636±0.007 | 0.639±0.008 | 0.405±0.014 | 0.517±0.010 | 0.439±0.012 | 0.265±0.012 | 0.354±0.014 | 0.317±0.013 |
| | S²GC | 0.835±0.002 | 0.569±0.007 | 0.664±0.007 | 0.661±0.007 | 0.422±0.010 | 0.535±0.010 | 0.454±0.011 | 0.279±0.014 | 0.366±0.014 | 0.320±0.013 |
| | GCE | 0.819±0.004 | 0.573±0.011 | 0.652±0.008 | 0.650±0.014 | 0.449±0.011 | 0.509±0.011 | 0.445±0.015 | 0.280±0.013 | 0.353±0.013 | 0.325±0.015 |
| | UnionNET | 0.820±0.006 | 0.569±0.014 | 0.664±0.007 | 0.653±0.012 | 0.452±0.010 | 0.541±0.010 | 0.450±0.009 | 0.283±0.014 | 0.370±0.011 | 0.320±0.012 |
| | NRGNN | 0.822±0.006 | 0.571±0.019 | 0.676±0.007 | 0.645±0.012 | 0.470±0.014 | 0.548±0.014 | 0.451±0.011 | 0.282±0.022 | 0.373±0.012 | 0.326±0.010 |
| | RTGNN | 0.828±0.003 | 0.570±0.010 | 0.682±0.008 | 0.678±0.011 | 0.474±0.011 | 0.555±0.010 | 0.457±0.009 | 0.280±0.011 | 0.386±0.014 | 0.342±0.016 |
| | SUGRL | 0.834±0.005 | 0.564±0.011 | 0.674±0.012 | 0.675±0.009 | 0.468±0.011 | 0.552±0.011 | 0.452±0.012 | 0.280±0.012 | 0.381±0.012 | 0.338±0.014 |
| | MERIT | 0.831±0.005 | 0.560±0.008 | 0.670±0.008 | 0.671±0.009 | 0.467±0.013 | 0.547±0.013 | 0.450±0.014 | 0.277±0.013 | 0.385±0.013 | 0.335±0.009 |
| | ARIEL | 0.843±0.004 | 0.573±0.013 | 0.681±0.010 | 0.675±0.009 | 0.471±0.012 | 0.553±0.012 | 0.455±0.014 | 0.284±0.014 | 0.389±0.013 | 0.343±0.013 |
| | SFA | 0.839±0.010 | 0.564±0.011 | 0.677±0.013 | 0.676±0.015 | 0.473±0.014 | 0.549±0.014 | 0.457±0.011 | 0.282±0.016 | 0.389±0.013 | 0.344±0.017 |
| | Sel-Cl | 0.828±0.002 | 0.570±0.010 | 0.685±0.012 | 0.676±0.009 | 0.472±0.013 | 0.554±0.014 | 0.455±0.011 | 0.282±0.017 | 0.389±0.013 | 0.341±0.015 |
| | Jo-SRC | 0.825±0.005 | 0.571±0.006 | 0.684±0.013 | 0.679±0.007 | 0.473±0.011 | 0.556±0.008 | 0.458±0.012 | 0.285±0.013 | 0.387±0.018 | 0.345±0.018 |
| | GRAND+ | 0.858±0.006 | 0.570±0.009 | 0.682±0.007 | 0.678±0.011 | 0.472±0.010 | 0.554±0.008 | 0.456±0.012 | 0.284±0.015 | 0.387±0.015 | 0.345±0.013 |
| | GFSA | 0.837±0.004 | 0.567±0.010 | 0.672±0.009 | 0.667±0.010 | 0.463±0.012 | 0.543±0.011 | 0.453±0.012 | 0.281±0.014 | 0.376±0.013 | 0.333±0.014 |
| | HONGAT | 0.841±0.005 | 0.571±0.008 | 0.678±0.011 | 0.673±0.012 | 0.469±0.013 | 0.551±0.012 | 0.456±0.011 | 0.283±0.015 | 0.384±0.014 | 0.340±0.015 |
| | CRGNN | 0.846±0.003 | 0.572±0.009 | 0.680±0.008 | 0.677±0.009 | 0.471±0.011 | 0.553±0.010 | 0.457±0.010 | 0.284±0.013 | 0.388±0.012 | 0.342±0.012 |
| | CGNN | 0.844±0.004 | 0.569±0.011 | 0.675±0.010 | 0.670±0.011 | 0.466±0.012 | 0.548±0.011 | 0.454±0.011 | 0.282±0.014 | 0.382±0.013 | 0.337±0.014 |
| | LR-GCL | 0.858±0.006 | 0.589±0.011 | 0.713±0.007 | 0.695±0.011 | 0.492±0.011 | 0.587±0.013 | 0.477±0.012 | 0.306±0.012 | 0.419±0.012 | 0.363±0.011 |
| | LRA-LR-GCL | 0.861±0.006 | 0.602±0.011 | 0.724±0.007 | 0.708±0.011 | 0.510±0.011 | 0.605±0.013 | 0.492±0.012 | 0.329±0.012 | 0.436±0.012 | 0.382±0.011 |
| Amazon-Photos | GCN | 0.703±0.005 | 0.475±0.023 | 0.501±0.013 | 0.529±0.009 | 0.351±0.014 | 0.341±0.014 | 0.372±0.011 | 0.291±0.022 | 0.281±0.019 | 0.290±0.014 |
| | S²GC | 0.736±0.005 | 0.488±0.013 | 0.528±0.013 | 0.553±0.008 | 0.363±0.012 | 0.367±0.014 | 0.390±0.013 | 0.304±0.024 | 0.284±0.019 | 0.288±0.011 |
| | GCE | 0.705±0.004 | 0.490±0.016 | 0.512±0.014 | 0.540±0.014 | 0.362±0.015 | 0.352±0.010 | 0.381±0.009 | 0.309±0.012 | 0.285±0.014 | 0.285±0.011 |
| | UnionNET | 0.706±0.006 | 0.499±0.015 | 0.547±0.014 | 0.545±0.013 | 0.379±0.013 | 0.399±0.013 | 0.379±0.012 | 0.322±0.021 | 0.302±0.013 | 0.290±0.012 |
| | NRGNN | 0.710±0.006 | 0.498±0.015 | 0.546±0.015 | 0.538±0.011 | 0.382±0.016 | 0.412±0.016 | 0.377±0.012 | 0.336±0.021 | 0.309±0.018 | 0.284±0.009 |
| | RTGNN | 0.746±0.008 | 0.498±0.007 | 0.556±0.007 | 0.550±0.012 | 0.392±0.010 | 0.424±0.013 | 0.390±0.014 | 0.348±0.017 | 0.308±0.016 | 0.302±0.011 |
| | SUGRL | 0.730±0.005 | 0.493±0.011 | 0.541±0.011 | 0.544±0.010 | 0.376±0.009 | 0.421±0.009 | 0.388±0.009 | 0.339±0.010 | 0.305±0.010 | 0.300±0.009 |
| | MERIT | 0.740±0.007 | 0.496±0.012 | 0.536±0.012 | 0.542±0.010 | 0.383±0.011 | 0.425±0.011 | 0.387±0.008 | 0.344±0.014 | 0.301±0.014 | 0.295±0.009 |
| | SFA | 0.740±0.011 | 0.502±0.014 | 0.532±0.015 | 0.547±0.013 | 0.390±0.014 | 0.433±0.014 | 0.389±0.012 | 0.347±0.016 | 0.312±0.015 | 0.299±0.013 |
| | ARIEL | 0.727±0.007 | 0.500±0.008 | 0.550±0.013 | 0.548±0.008 | 0.391±0.009 | 0.427±0.012 | 0.389±0.014 | 0.349±0.014 | 0.307±0.013 | 0.299±0.013 |
| | Sel-Cl | 0.725±0.008 | 0.499±0.012 | 0.551±0.010 | 0.549±0.008 | 0.389±0.011 | 0.426±0.008 | 0.391±0.020 | 0.350±0.018 | 0.310±0.015 | 0.300±0.017 |
| | Jo-SRC | 0.730±0.005 | 0.500±0.013 | 0.555±0.011 | 0.551±0.011 | 0.394±0.013 | 0.425±0.013 | 0.393±0.013 | 0.351±0.013 | 0.305±0.018 | 0.303±0.013 |
| | GRAND+ | 0.756±0.004 | 0.497±0.010 | 0.553±0.010 | 0.552±0.011 | 0.390±0.013 | 0.422±0.013 | 0.387±0.013 | 0.348±0.013 | 0.309±0.014 | 0.302±0.012 |
| | GFSA | 0.722±0.006 | 0.492±0.012 | 0.530±0.012 | 0.543±0.010 | 0.372±0.012 | 0.398±0.011 | 0.383±0.011 | 0.328±0.016 | 0.294±0.015 | 0.292±0.012 |
| | HONGAT | 0.738±0.005 | 0.496±0.010 | 0.542±0.011 | 0.547±0.009 | 0.384±0.013 | 0.415±0.012 | 0.388±0.012 | 0.342±0.015 | 0.303±0.014 | 0.298±0.011 |
| | CRGNN | 0.744±0.004 | 0.501±0.009 | 0.548±0.010 | 0.549±0.008 | 0.388±0.011 | 0.422±0.010 | 0.390±0.010 | 0.346±0.014 | 0.306±0.013 | 0.301±0.010 |
| | CGNN | 0.732±0.007 | 0.494±0.011 | 0.538±0.013 | 0.545±0.011 | 0.378±0.012 | 0.408±0.013 | 0.385±0.013 | 0.335±0.017 | 0.300±0.016 | 0.296±0.013 |
| | LR-GCL | 0.757±0.010 | 0.520±0.013 | 0.581±0.013 | 0.570±0.007 | 0.410±0.014 | 0.455±0.014 | 0.406±0.012 | 0.369±0.012 | 0.335±0.014 | 0.318±0.010 |
| | LRA-LR-GCL | 0.762±0.010 | 0.533±0.013 | 0.597±0.013 | 0.588±0.007 | 0.430±0.014 | 0.472±0.014 | 0.423±0.012 | 0.392±0.012 | 0.352±0.014 | 0.325±0.010 |

Table 9: Training time (seconds) comparisons for node classification.

| Methods | Cora | Citeseer | PubMed | Coauthor-CS | Wiki-CS | Computer | Photo | ogbn-arxiv |
|---------|------|----------|--------|-------------|---------|----------|-------|------------|
| GCN | 11.5 | 13.7 | 38.6 | 43.2 | 22.3 | 30.2 | 19.0 | 215.1 |
| S$^2$GC | 20.7 | 22.5 | 47.2 | 57.2 | 27.6 | 38.5 | 22.2 | 243.7 |
| GCE | 32.6 | 36.9 | 67.3 | 80.8 | 37.6 | 50.1 | 32.2 | 346.1 |
| UnionNET | 67.5 | 69.7 | 100.5 | 124.2 | 53.2 | 69.2 | 45.3 | 479.3 |
| NRGNN | 72.4 | 80.5 | 142.7 | 189.4 | 74.3 | 97.2 | 62.4 | 650.2 |
| RTGNN | 143.3 | 169.5 | 299.5 | 353.5 | 153.7 | 201.5 | 124.2 | 1322.2 |
| SUGRL | 100.3 | 122.1 | 207.4 | 227.1 | 107.7 | 142.8 | 87.7 | 946.8 |
| MERIT | 167.2 | 179.2 | 336.7 | 375.3 | 172.3 | 226.5 | 140.6 | 1495.1 |
| ARIEL | 156.9 | 164.3 | 284.3 | 332.6 | 145.1 | 190.4 | 118.3 | 1261.4 |
| SFA | 237.5 | 269.4 | 457.1 | 492.3 | 233.5 | 304.5 | 187.2 | 2013.1 |
| Sel-Cl | 177.3 | 189.9 | 313.5 | 352.5 | 161.7 | 211.1 | 130.9 | 1401.1 |
| Jo-SRC | 148.2 | 157.1 | 281.0 | 306.1 | 144.5 | 188.0 | 118.5 | 1256.0 |
| GRAND+ | 57.4 | 68.4 | 101.7 | 124.2 | 54.8 | 73.8 | 44.5 | 479.2 |
| LR-GCL | 159.9 | 174.5 | 350.7 | 380.9 | 180.3 | 235.7 | 145.5 | 1552.7 |

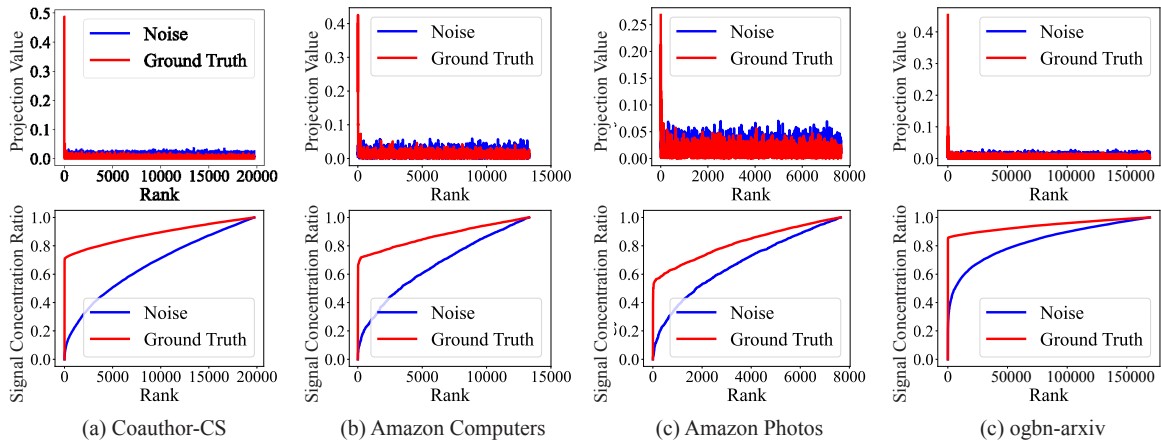

(a) Coauthor-CS   (b) Amazon Computers   (c) Amazon Photos   (c) ogbn-arxiv

Figure 3: Eigen-projection (first row) and energy concentration (second row) on Coauthor-CS, Amazon-Computers, Amazon-Photos, and ogbn-arxiv. By the rank of $0.2 \min \{N, d\}$, the concentration entropy on Coauthor-CS, Amazon-Computers, Amazon-Photos, and ogbn-arxiv are 0.779, 0.809, 0.752, and 0.787.

in Section 5.3. We adopt TEDGCN (Yan et al., 2023), which is a widely used GNN for semi-supervised node classification on heterophilic graphs, as the GNN encoder in LR-GCL and LRA-LR-GCL. The results are shown in Table 10. It is observed that LR-GCL and LRA-LR-GCL show significantly improved performance over the heterophilic GNN for semi-supervised node classification with the presence of different types of noise.

Table 10: Performance comparison for node classification on Cora, Citeseer, PubMed, and Wiki-CS with asymmetric label noise, symmetric label noise, and attribute noise.

| Dataset | Methods | Noise Type | | | | | | | | | |
|---------|---------|---|---|---|---|---|---|---|---|---|---|
| | | 0 | 40 | | | 60 | | | 80 | | |
| | | - | Asymmetric | Symmetric | Attribute | Asymmetric | Symmetric | Attribute | Asymmetric | Symmetric | Attribute |
| Texas | TEDGCN | 0.771±0.025 | 0.525±0.023 | 0.528±0.018 | 0.541±0.022 | 0.402±0.016 | 0.418±0.019 | 0.445±0.021 | 0.312±0.015 | 0.328±0.017 | 0.341±0.020 |
| | LR-GCL | 0.780±0.013 | 0.547±0.019 | 0.557±0.016 | 0.568±0.017 | 0.438±0.015 | 0.444±0.017 | 0.463±0.018 | 0.336±0.012 | 0.353±0.014 | 0.365±0.016 |
| | LRA-LR-GCL | **0.785±0.018** | **0.556±0.016** | **0.563±0.013** | **0.576±0.015** | **0.451±0.012** | **0.452±0.014** | **0.472±0.016** | **0.338±0.010** | **0.367±0.012** | **0.372±0.014** |
| Chameleon | TEDGCN | 0.569±0.009 | 0.382±0.021 | 0.401±0.018 | 0.425±0.020 | 0.298±0.017 | 0.315±0.019 | 0.328±0.022 | 0.225±0.016 | 0.241±0.018 | 0.254±0.021 |
| | LR-GCL | 0.584±0.011 | 0.407±0.019 | 0.436±0.015 | 0.447±0.018 | 0.332±0.015 | 0.342±0.016 | 0.356±0.018 | 0.251±0.013 | 0.269±0.015 | 0.283±0.017 |
| | LRA-LR-GCL | **0.585±0.008** | **0.412±0.016** | **0.444±0.013** | **0.452±0.014** | **0.341±0.011** | **0.352±0.013** | **0.361±0.015** | **0.262±0.010** | **0.282±0.012** | **0.290±0.014** |

## C   Additional Implementation Details

Jo-SRC utilizes the Jensen-Shannon divergence to identify clean training samples through a general representation space selection strategy. This approach also incorporates a consistency regularization term into

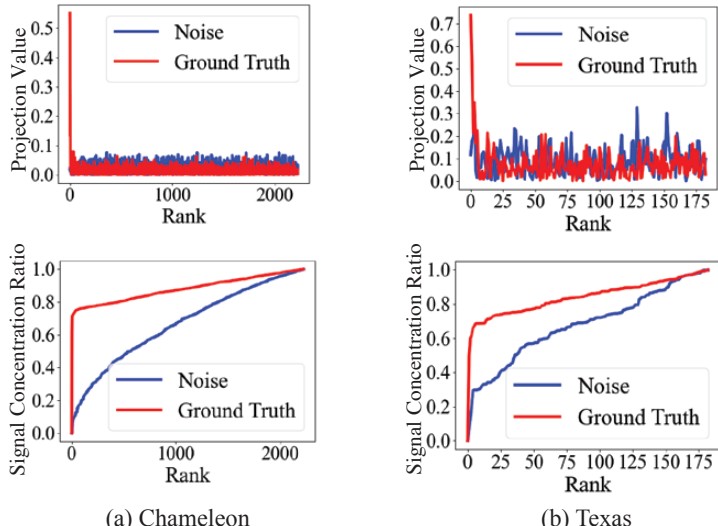

Figure 4: Eigen-projection (first row) and signal concentration ratio (second row) on Chameleon and Texas. The study in this figure is performed for asymmetric label noise with a noise level of 60%. By the rank of $0.2 \min \{N, d\}$, the concentration entropy on Chameleon and Texas are 0.762 and 0.725.

the contrastive loss to enhance robustness. In our adaptation, we apply the sample selection and consistency regularization techniques in Jo-SRC to the state-of-the-art GCL method, MERIT. We modify the graph contrastive loss to integrate the regularization term from Jo-SRC and train the GCL encoder exclusively on the clean samples identified by Jo-SRC.

Sel-CL is designed to learn robust pre-trained representations by selectively forming contrastive pairs from confident examples. These confident examples are identified through the alignment of learned representations with propagated labels, assessed using cross-entropy loss. Sel-CL then selects contrastive pairs that exhibit a representation similarity exceeding a dynamically determined threshold. We adopt the confident contrastive pair selection strategy in Sel-CL to select the confident contrastive pairs in the node representation space. The selection strategy is incorporated into the state-of-the-art GCL method, MERIT.

