# OpenReview forum: "Low-Rank Graph Contrastive Learning for Noisy Node Classification"
_TMLR — Rejected by TMLR_

### Review · Reviewer_ANvn · 2025-02-21

**Summary Of Contributions:**

This paper identifies the low-frequency property of the ground truth label through empirical study. Inspired by this discovery, the authors propose Low-Rank Graph Contrastive Learning (LR-GCL), which incorporates the truncated nuclear norm as a low-rank regularization term in the loss function of standard prototypical graph contrastive learning. Theoretical analysis and extensive experiments on public benchmarks demonstrate LR-GCL's superior performance and robustness in learning node representations.

**Audience:**

Yes

**Broader Impact Concerns:**

Given the scope of this paper, I don’t think there are any ethical concerns.

**Claims And Evidence:**

Yes

**Requested Changes:**

1. (Required) Address my concerns above.
2. (Enhancement) The article is not well organized and needs adjustment, here are my suggestions:
    1. Figure 1 is referred to in Section 1.1 before the notation description in Section 3.1. However, the caption of Figure 1 is full of notations, which may increase the reader's comprehension burden and require unnecessary paragraphs to explain the role of these notations. Consider moving the empirical study from Section 1.1 to Section 4 before the method description. This way, all notations are predefined, and the explanation can naturally and logically lead to the subsequent LR-GCL method.
    2. The node-wise contrastive loss and the prototypical contrastive loss in Section 4.1 can be introduced separately. The motivation for each loss should be described in detail. For instance, the authors should explain how these designs help address feature noise and label noise.
    3. The introduction section should be re-organized. Currently, there are many works that address label noise or feature noise problems on graphs. The authors should focus on discussing the issues with these existing methods and how their work aims to solve these problems, rather than going into too much detail about the proposed LR-GCL method in the introduction section.

3. (Enhancement) Add necessary citations:
    1. In section 1 and section 2.1, the author claimed that noise can propagate through graph topology. This is an intuitive conclusion, but the author still needs to add the necessary citations, for example:
        1. Dai, Enyan et al. "Nrgnn: Learning a label noise resistant graph neural network on sparsely and noisily labeled graphs." Proceedings of the 27th ACM SIGKDD conference on knowledge discovery & data mining. 2021.  (In section 2.2 “noisy information can propagate via message passing of GNNs.”)
        2. Zhonghao, Wang et al. "NoisyGL: A Comprehensive Benchmark for Graph Neural Networks under Label Noise" Advances in Neural Information Processing Systems 37, (2025). (In section 4 “Graph structure can amplify the negative effect of label noise.”)
        3. Wang, Botao, et al. "Deep insights into noisy pseudo labeling on graph data." Advances in Neural Information Processing Systems 36 (2024). (In section I “the mislabeled nodes can inappropriately reduce the predicted confidence of their 2-hop neighbors.”)
    2. In section 1, the author claimed “In fact, noise inherently exists in the graph data for many real-world applications. Such noise may be present in node attributes or node labels, which forms two types of noise, attribute noise and label noise.” Add citation here.

4.  (Enhancement) Suggested minor modifications:
    1. In Figure 1, the author should specify the type of label noise injected into the graph data: symmetric (uniform) noise or asymmetric (pair) noise.
    2. The detailed information about the baseline methods Jo-SRC and Sel-CL in Section 5.3 can be moved to the appendix.
    3. In Section 5.1, the author should describe the data splitting strategy (train/test/valid). Additionally, they should discuss important experiment details, such as the fact that the training and validation sets consist of noisy data, while the clean data in the test set is used to evaluate performance.
    4. In Section 5.1, the authors should formally define symmetric and asymmetric noise and provide a citation, for example:
        1. Song, Hwanjun, et al. "Learning from noisy labels with deep neural networks: A survey." IEEE transactions on neural networks and learning systems 34.11 (2022): 8135-8153. (In Section II.B Taxonomy of Label Noise, see the definition of instance-independent Label Noise)
    5. In the first line of Table 7, "noise type" is a more suitable term than "noise level."
    6. The y-axis label of the second line in Figure 1 is "energy concentration," which does not correspond with the caption description "signal concentration ratio." Please update either the label or the caption to ensure consistency.
    7. The description of 𝐻 throughout the article is not consistent. In Section 3.2, 𝐻 is defined as the node representation generated by GCN. However, in Section 4.2, 𝐻 is described as "low-rank features." The authors should ensure consistent usage and description of notations.

**Strengths And Weaknesses:**

Pros:
1. The general motivation of this paper is clear. The authors demonstrated the low rank property of the ground truth label, and the proposed solution is adding a low-rank regularization term into loss function.
2. The research scope of this paper is relatively novel. While numerous studies focus on tackling label noise, feature noise, and topology noise individually, only a few attempts to handle two or more simultaneously. In real-world applications, these three types of noise can coexist on graph data. Therefore, tackling both label noise and feature noise is an interesting and important problem.

Cons:
1. For contrastive learning, the role of contrastive learning in dealing with label or attribute noise is not thoroughly discussed. While the authors provide a motivation for using low-rank representation to address the label noise problem through empirical study, contrastive learning seems more like a performance-enhancing trick. Although Section 4.1 mentions, "It is preferred that the mutual information between H1 and H2 is maximized," this is not sufficient to demonstrate the effectiveness of contrastive learning in graph learning with noisy data. Additionally, previous works have employed contrastive strategies in graph learning with label noise, such as:

&emsp; a) Li, Xianxian, et al. "Contrastive learning of graphs under label noise." Neural Networks 172 (2024): 106113.

&emsp; b) Yuan, Jingyang, et al. "Learning on graphs under label noise." ICASSP 2023-2023 IEEE International Conference on Acoustics, Speech and Signal Processing (ICASSP). IEEE, 2023.

The authors have not discussed the improvement or differences between these methods and the proposed LR-GCL. Additionally, these methods should be included in the baseline comparisons.

2. For the low-rank regularization, it is designed to enhance the low-pass filtering property of GNNs. However, previous studies have shown that prevalent GNNs are inherently low-pass filters, as demonstrated by:

&emsp; a) Nt, Hoang, and Takanori Maehara. "Revisiting graph neural networks: All we have is low-pass filters." arXiv preprint arXiv:1905.09550 (2019).

Therefore, my first concern is what’s the difference between the low-pass filtering of GNN and that of LR-GCL? Secondly, this approach may not be suitable for graph data with high-frequency signals, such as heterophily graphs. However, the datasets selected in this work are all homophily. The authors have not discussed these issues.

3. For the theoretical analysis, Theorem 4.1 argues that using low-rank features to minimize empirical risk can minimize expected risk. However, this theorem is not directly related to label noise or feature noise. The authors should prove that minimizing empirical risk on noisy data can minimize expected risk on clean data. For an ideal theoretical analysis, please refer to Theorems 1, 2, and 3 in the following paper:

&emsp; a) Ghosh, Aritra, Himanshu Kumar, and P. Shanti Sastry. "Robust loss functions under label noise for deep neural networks." Proceedings of the AAAI conference on artificial intelligence. Vol. 31. No. 1. 2017.

---

> ### Author Response · Authors · 2025-04-12
> **Response to Reviewer ANvn Part 1**
>
> We appreciate the review and the suggestions in this review. The raised issues are addressed below.
>
> **Responses to the Cons**
>
> **1. “...the role of contrastive learning in dealing with label or attribute noise is not thoroughly discussed...”**
>
> We respectfully disagree that contrastive learning in LR-GCL is a performance-enhancing trick. Instead, contrastive learning plays a crucial role in enhancing the robustness of node representations in the presence of label and attribute noise. The literature on contrastive learning has demonstrated its effectiveness in combating label noise and attribute noise [1,2,3,4], benefiting from its learning objective of instance discrimination instead of solely relying on potentially corrupted labels and attributes. The graph contrastive learning architecture has also demonstrated its effectiveness in learning robust node representations against different kinds of noise in the node attributes and the node labels [5, 6, 7, 8, 9].
>
> The major novelty and difference of our LR-GCL compared to existing GCL works, such as CRGNN [9] and CGNN [10],  is that the low-rank learning is explicitly enforced in the training process of the GCL encoder by adding the Truncated Nuclear Norm (TNN) of the gram matrix of the node features to the training loss (Equation  (2)) of the LR-GCL encoder.  By optimizing such training loss of the LR-GCL encoder, our LR-GCL exhibits much lower Kernel Complexity (KC) compared to the competing methods, as shown in Table 5 and Table 6 of our revised paper. We also provided a theoretical guarantee in Theorem 4.1 on the upper bound for the test loss of a linear classifier trained on the node representations. A lower TNN leads to a lower KC, contributing to a lower upper bound for the test loss, which theoretically justifies the low-rank learning by optimization of the TNN. Furthermore, inspired by the generalization bound for the test loss in Theorem 4.1, we propose LRA-LR-GCL, which further reduces the kernel complexity of LR-GCL. Extended experimental results in Table 3 and Table 8 of the revised paper show that LRA-LR-GCL even outperforms LR-GCL with noisy labels and noisy attributes, and Table 5 and Table 6 of the revised paper show that LRA-LR-GCL exhibits lower KC and lower upper bound for the test loss than that of LR-GCL.
>
> In addition, to demonstrate the effectiveness of LR-GCL, we compare LR-GCL with CRGNN [9] and CGNN [10] for node classification with label noise and attribute noise. The results are shown in the table below, where LR-GCL outperforms all the baselines by a large margin, especially for more noisy labels and more noisy attributes.
>
> Performance comparison for node classification on Cora, Citeseer, and Pubmed with asymmetric label noise, symmetric label noise, and attribute noise. Comparisons on Coauthor-CS, ogbn-arxiv, Wiki-CS, Amazon-Computers, and Amazon-Photos are shown in Table 3 and Table 8 of our revised paper.
> |Dataset|Methods|Clean|Asymmetric (40)|Symmetric (40)|Attribute (40)|Asymmetric (60)|Symmetric (60)|Attribute (60)|Asymmetric (80)|Symmetric (80)|Attribute (80)|
> |:-:|:-:|:-:|:-:|:-:|:-:|:-:|:-:|:-:|:-:|:-:|:-:|
> |Cora|CRGNN[9]|0.842|0.572|0.678|0.674|0.470|0.551|0.454|0.283|0.386|0.341|
> |Cora|CGNN[10]|0.835|0.567|0.670|0.669|0.462|0.544|0.450|0.281|0.380|0.337|
> |Cora|LR-GCL|0.858|0.589|0.713|0.695|0.492|0.587|0.477|0.306|0.419|0.363|
> |Citeseer|CRGNN[9]|0.751|0.497|0.552|0.549|0.389|0.423|0.388|0.347|0.310|0.301|
> |Citeseer|CGNN[10]|0.741|0.493|0.544|0.546|0.385|0.419|0.385|0.343|0.307|0.297|
> |Citeseer|LR-GCL|0.757|0.520|0.581|0.570|0.410|0.455|0.406|0.369|0.335|0.318|
> |Pubmed|CRGNN[9]|0.829|0.612|0.623|0.618|0.452|0.455|0.503|0.335|0.457|0.457|
> |Pubmed|CGNN[10]|0.822|0.607|0.620|0.615|0.449|0.451|0.499|0.332|0.454|0.454|
> |Pubmed|LR-GCL|0.845|0.637|0.645|0.637|0.479|0.484|0.526|0.356|0.482|0.482|

---

> ### Author Response · Authors · 2025-04-12
> **Response to Reviewer ANvn Part 2**
>
> **2. “...what’s the difference between the low-pass filtering of GNN and that of LR-GCL?...this approach may not be suitable for graph data with high-frequency signals, such as heterophily graphs...”**
>
>
> Although GNNs are considered low-pass filtering, the effect of such low-pass filtering is not strong enough to capture the Low-Frequency Property (LFP), which is introduced in Figure 2, Figure 3, and Figure 4 of the revised paper, in the noisy labels. In contrast, our LR-GCL better captures the LFP in the noisy labels by learning low-rank features. We remark that low-rank learning exhibits superior performance for noisy attributes in [13] through learnable low-rank filters. Moreover, recent works on graph attention/transformers have shown that finding a good balance between low-frequency and high-frequency information in the graph benefits node representation learning for graph learning tasks such as node classification [11, 12]. Compared with the existing GNNs and graph attention/transformer methods, our LR-GCL learns a better balance between low-frequency and high-frequency information, with more focus on the low-frequency part by minimizing the truncated nuclear norm due to LFP. As shown in Table 3 of our revised paper, LR-GCL exhibits better node classification accuracy than the graph attention/transformer methods, GFSA [11] and HONGAT [12], when label noise or attribute noise is present in the input graph. In addition, the balance between the low-frequency and high-frequency information can be quantitatively measured by the kernel complexity defined in Section 4.2. As shown in Table 6 of our revised paper, the node representations learned by LR-GCL exhibit lower kernel complexity than those of graph contrastive learning methods and graph attention/transformer methods.
>
> We herein provide evidence that LFP still exists in heterophilic graphs. To study the LFP on heterophilic graphs, we compute the signal concentration ratio (defined in the caption of Figure 2) of two widely used heterophilic graph datasets, Chameleon and Texas [14], following the way we demonstrate LFP on homophilic graphs in Figure 2 and Figure 3. By using a rank of $r=0.2\min \lbrace N, d \rbrace$, the signal concentration ratios of Chameleon and Texas are 0.762 and 0.725, respectively, which are close to what we have observed on homophilic graphs in Figure 3 with the same rank. These results show that LFP also exists in heterophilic graph data.
>
> Moreover, we apply LR-GCL on public heterophilic graph benchmarks, Film and Chameleon [14], to validate the effectiveness of LR-GCL. The results in the table below show that LR-GCL significantly outperforms other robust learning methods on the heterophilic graphs. We adopt TEDGCN [15], which is a widely used GNN for semi-supervised node classification on heterophilic graphs, as the GNN encoder in LR-GCL and LRA-LR-GCL. The results are shown in the table below. It is observed that LR-GCL and LRA-LR-GCL show significantly improved performance over the heterophilic GNN for semi-supervised node classification with the presence of different types of noise
>
>
> Performance comparison for node classification on Texas and Chameleon with asymmetric label noise, symmetric label noise, and attribute noise. The standard deviation for the results is shown in Table 10 of the revised paper.
>
> |Dataset|Methods|Clean|Asymmetric (40)|Symmetric (40)|Attribute (40)|Asymmetric (60)|Symmetric (60)|Attribute (60)|Asymmetric (80)|Symmetric (80)|Attribute (80)|
> |:-:|:-:|:-:|:-:|:-:|:-:|:-:|:-:|:-:|:-:|:-:|:-:|
> | Texas     | TEDGCN        | 0.771 | 0.525    | 0.528   | 0.541    | 0.402    | 0.418   | 0.445    | 0.312    | 0.328   | 0.341    |
> | Texas     | LR-GCL        | 0.780 | 0.547    | 0.557   | 0.568    | 0.438    | 0.444   | 0.463    | 0.336    | 0.353   | 0.365    |
> | Texas     | LRA-LR-GCL    | 0.785 | 0.556    | 0.563   | 0.576    | 0.452    | 0.472   | 0.486    | 0.338    | 0.367   | 0.372    |
> | Chameleon | TEDGCN        | 0.569 | 0.382    | 0.401   | 0.398    | 0.298    | 0.315   | 0.328    | 0.225    | 0.241   | 0.254    |
> | Chameleon | LR-GCL        | 0.584 | 0.407    | 0.426   | 0.427    | 0.332    | 0.342   | 0.365    | 0.251    | 0.269   | 0.283    |
> | Chameleon | LRA-LR-GCL    | 0.585 | 0.412    | 0.444   | 0.452    | 0.352    | 0.361   | 0.365    | 0.262    | 0.282   | 0.290    |

---

> ### Author Response · Authors · 2025-04-12
> **Response to Reviewer ANvn Part 3**
>
> **3. “...this theorem is not directly related to label noise or feature noise...”**
>
> We have improved our theoretical result in Theorem 4.1 of the revised paper. The test loss in Theorem 4.1 now measures the $\ell^2$-distance between the output of a linear classifier trained on data with noisy label $\mathbf Y$, and the ground truth clean label $\tilde{\mathbf Y}$, in response to the request in this review. We remark that such  $\ell^2$-distance between classifier output and the ground truth clean label is close to the “noise-tolerant” effect defined in [16] as mentioned in this review. Please also note that LR-GCL has a different scope from that of [16]: [16] focuses on loss functions robust to label noise, while LR-GCL focuses on low-rank learning for mitigating the adverse effect of the label noise (it is also empirically robust to attribute noise). Equation (4)  in Theorem 4.1 of the revised paper shows that the upper bound for the test loss of a linear classifier trained on the node representations, or the generalization bound, for transductive node classification comprises the following three quantities, $L\_1(\mathbf{K}, \mathbf{\tilde{Y}}, t)$, $L\_2(\mathbf{K}, \mathbf{N}, t)$, and $\text{KC}({\mathbf{K}})$, which are explained as follows and also in the revised paper. $\mathbf{K}$ is the kernel gram matrix of the node representations $\mathbf{H}$ generated by the LR-GCL encoder. $\mathbf{\tilde{Y}}$ is the ground-truth clean label. $t$ is the epoch number. $\mathbf{N}$ is the label noise.
>
> - $L\_1(\mathbf{K}, \mathbf{\tilde{Y}}, t)$ corresponds to the training loss of the node classifier with the clean label.
>
> - $L\_2(\mathbf{K}, \mathbf{N}, t)$ corresponds to the loss incurred by the label noise.
>
> - $\text{KC}({\mathbf{K}})$ is the kernel complexity (KC), which measures the complexity of the kernel gram matrix $\mathbf{K} =\mathbf{H} \mathbf{H}^{\top}$ computed by the node representation $\mathbf H$.
>
> The KC $\text{KC}({\mathbf{K}})$ is computed using the truncated nuclear norm of the gram matrix of the node representations, $||\mathbf{K}||\_{r\_0}$, which is explicitly added to the training loss of our LR-GCL encoder in Equation (2). A lower $||\mathbf{K}||\_{r\_0}$ leads to a lower $\text{KC}({\mathbf{K}})$ contributing to a lower generalization bound, which justifies why we learn the low-rank features $\mathbf{H}$ in LR-GCL by adding the TNN $||\mathbf{K}||\_{r\_0}$ to the training loss of our LR-GCL encoder. A smaller $||\mathbf{K}||\_{r\_0}$ is obtained by optimizing such training loss, which in turn ensures a smaller KC contributing to a smaller generalization bound for transductive node classification, which is empirically observed in Table 5 and Table 6 of the revised paper.
>
> For your convenience, the comparison on $L\_1(\mathbf{K}, \mathbf{\tilde{Y}}, t)$, $L\_2(\mathbf{K}, \mathbf{N}, t)$, $\text{KC}({\mathbf{K}})$, and the upper bound for the test loss between LR-GCL, LRA-LR-GCL, and the competing baseline methods are shown in the table below (also Table 6 of the revised paper). It is observed that LRA-LR-GCL and  LR-GCL outperform all the other competing baselines with lower KC and lower upper bound for the test loss, demonstrating the effectiveness of our low-rank learning methods which are inspired by LFP and our generalization bound in Theorem 4.1.
>
> |Datasets|Metric|MERIT|SFA|Jo-SRC|GCN|GFSA|HONGAT|LR-GCL|LRA-LR-GCL|
> |:-:|:-:|:-:|:-:|:-:|:-:|:-:|:-:|:-:|:-:|
> |Cora|$L\_1$|5.24|6.04|6.50|7.38|6.44|6.38|3.72|**3.65**|
> ||$L\_2$|4.92|4.95|5.05|5.24|3.80|4.25|2.97|**2.72**|
> ||KC|0.37|0.42|0.48|0.44|0.35|0.40|0.20|**0.18**|
> ||Upper Bound|10.68|11.59|12.18|13.22|10.80|11.25|7.05|**6.74**|
> |Citeseer|$L\_1$|4.72|4.85|4.92|5.10|4.54|4.69|4.02|**3.95**|
> ||$L\_2$|4.33|4.69|4.42|5.08|4.20|4.42|3.75|**3.60**|
> ||KC|0.47|0.45|0.55|0.64|0.47|0.50|0.24|**0.21**|
> ||Upper Bound|9.77|10.21|10.17|11.07|9.40|9.84|8.20|**7.97**|
> |PubMed|$L\_1$|3.97|4.02|4.11|4.35|4.26|3.95|3.38|**3.40**|
> ||$L\_2$|2.69|2.54|2.60|2.88|2.98|2.85|2.32|**2.26**|
> ||KC|0.54|0.50|0.62|0.71|0.52|0.66|0.30|**0.28**|
> ||Upper Bound|7.44|7.28|7.59|8.15|7.99|7.63|6.25|**6.16**|

---

> > ### Author Response · Authors · 2025-04-12
> > **Response to Reviewer ANvn Part 4**
> >
> > **Responses to the Requested Changes**
> >
> >
> >
> > **1. “(Required) Address my concerns above.“**
> >
> > Please refer to our response to the weaknesses above.
> >
> > **2. “(Enhancement) The article is not well organized and needs adjustment...“**
> >
> >
> >
> > **(1). “Figure 1 is referred to in Section 1.1 before the notation description...Consider moving the empirical study from Section 1.1 to Section 4 before the method description....“**
> >
> > We have moved Figure 1 from Section 1.1 to Section 5.7 in the revised paper.
> >
> > **(2). “The node-wise contrastive loss and the prototypical contrastive loss in Section 4.1 can be introduced separately...explain how these designs help address feature noise and label noise.“**
> >
> > The discussion about the node-wise contrastive loss and the prototypical contrastive loss has been revised in Section 4.1 of the revised paper, which is also copied below for your convenience.
> >
> >
> > The node-wise contrastive loss encourages consistency between node representations across two perturbed views of the input graph. This design is particularly helpful in mitigating the impact of attribute noise, as the perturbations simulate different noise patterns. By maximizing agreement between representations from these views, the model learns to extract noise-invariant features that are robust to corruptions in input attributes.
> >
> > The prototypical contrastive loss clusters node representations and enforces alignment between individual nodes and their corresponding cluster prototypes. This helps address label noise by leveraging semantic consistency across nodes within the same cluster. Even if a node’s label is corrupted, the prototype, which is computed from a group of similar nodes in a cluster, provides a denoised supervisory signal that guides the representation toward its correct semantic class.
> >
> >
> > **(3). “The introduction section should be reorganized. Currently, there are many works that address label noise or feature noise problems on graphs....“**
> >
> > We have added a paragraph in the introduction which summarizes the existing works and why they are not sufficient to address the label noise and feature noise problems on graphs.
> >
> > **3. “(Enhancement) Add necessary citations...“**
> >
> > We have added the requested citations to the revised paper.
> >
> > **4. “(Enhancement) Suggested minor modifications...“**
> >
> > **(1) “In Figure 1, the author should specify the type of label noise injected into the graph data: symmetric (uniform) noise or asymmetric (pair) noise.“**
> >
> > The study in this figure (Figure 2 in the revised paper) is performed for asymmetric label noise with a noise level of $60\%$. We have added this information in the caption of Figure 2 in the revised paper.
> >
> >
> > **(2) “The detailed information about the baseline methods Jo-SRC and Sel-CL in Section 5.3 can be moved to the appendix.“**
> >
> > We have moved the detailed information about the baseline methods Jo-SRC and Sel-CL to Section C of the appendix in the revised paper.
> >
> > **(3) “In Section 5.1, the author should describe the data splitting strategy (train/test/valid)...“**
> >
> > We have added detailed information about the data splitting strategy and the fact that the training and validation sets consist of noisy data, while the clean data in the test set is used to evaluate performance in Section 5.1 of the revised paper.
> >
> > **(4) “In Section 5.1, the authors should formally define symmetric and asymmetric noise and provide a citation...“**
> >
> > We have added the formal definitions of symmetric and asymmetric noise and the suggested citation to Section 5.1 of the revised paper.
> >
> > **(5) “In the first line of Table 7, "noise type" is a more suitable term than "noise level.“**
> >
> > We have revised the term "noise level" to "noise type" in the revised paper.
> >
> > **(6) “The y-axis label of the second line in Figure 1 is "energy concentration," which does not correspond with the caption description "signal concentration ratio.“**
> >
> > We have revised the y-axis label of the second line in Figure 1 (Figure 2 in the revised paper) to "signal concentration ratio".
> >
> > **(7) “The description of  $\mathbf H$ throughout the article is not consistent...“**
> >
> > We have revised the notation $\mathbf H$ to $\hat{\mathbf H}$  in Section 3.2 of the revised paper so that it is distinct from the low-rank features $\mathbf H$ in Section 4.

---

> > > ### Author Response · Authors · 2025-04-12
> > > **References in the Response**
> > >
> > > [1] Xue, Yihao, et al. "Investigating why contrastive learning benefits robustness against label noise." International Conference on Machine Learning. PMLR, 2022.
> > >
> > > [2] Ghosh, Aritra, et al. "Contrastive learning improves model robustness under label noise." Proceedings of the IEEE/CVF conference on computer vision and pattern recognition. 2021.
> > >
> > > [3] Liu, Jingyi, et al. "DN-CL: Deep Symbolic Regression against Noise via Contrastive Learning." arXiv preprint arXiv:2406.14844 (2024).
> > >
> > > [4] Chuang, Ching-Yao, et al. "Robust contrastive learning against noisy views." Proceedings of the IEEE/CVF conference on computer vision and pattern recognition. 2022.
> > >
> > > [5] Zhang, Yifei, et al. "Spectral feature augmentation for graph contrastive learning and beyond." Proceedings of the AAAI conference on artificial intelligence. Vol. 37. No. 9. 2023.
> > >
> > > [6] Guo, Jiayan, et al. "Learning robust representation through graph adversarial contrastive learning." International Conference on Database Systems for Advanced Applications. Cham: Springer International Publishing, 2022.
> > >
> > > [7] Lin, Minhua, et al. "Certifiably robust graph contrastive learning." Advances in Neural Information Processing Systems 36 (2023): 17008-17037.
> > >
> > > [8] In, Yeonjun, et al. "Similarity preserving adversarial graph contrastive learning." Proceedings of the 29th ACM SIGKDD Conference on Knowledge Discovery and Data Mining. 2023.
> > >
> > > [9] Li, Xianxian, et al. "Contrastive learning of graphs under label noise." Neural Networks 172 (2024): 106113.
> > >
> > > [10] Yuan, Jingyang, et al. "Learning on graphs under label noise." ICASSP 2023-2023 IEEE International Conference on Acoustics, Speech and Signal Processing (ICASSP). IEEE, 2023.
> > >
> > > [11] Choi, Jeongwhan, et al. "Graph convolutions enrich the self-attention in transformers!." Advances in Neural Information Processing Systems 37 (2024): 52891-52936.
> > >
> > > [12] Zhang, Heng-Kai, et al. "HONGAT: graph attention networks in the presence of high-order neighbors." Proceedings of the AAAI Conference on Artificial Intelligence. Vol. 38. No. 15. 2024.
> > >
> > > [13] Cheng, Xiuyuan, et al. "Graph Convolution with Low-rank Learnable Local Filters." International Conference on Learning Representations. 2021.
> > >
> > > [14] Luan et al. "Revisiting heterophily for graph neural networks." NeurIPS 2022.
> > >
> > > [15] Yan, Yuchen, et al. "From trainable negative depth to edge heterophily in graphs." Advances in Neural Information Processing Systems 36 (2023): 70162-70178.
> > >
> > > [16] Ghosh, Aritra, et al. "Robust loss functions under label noise for deep neural networks." Proceedings of the AAAI conference on artificial intelligence (AAAI) 2017.

---

> > > > ### Comment · Reviewer_ANvn · 2025-04-21
> > > >
> > > > I appreciate the authors' diligent efforts in revising this manuscript. I have reviewed the revised version, I am pleased to confirm that all my previous concerns have been addressed.
> > > >
> > > > Typo in Section 1, the following sentence appears duplicated:
> > > > "While several methods (Dai et al., 2021; Qian et al., 2022; Zhuang & Al Hasan, 2022) have extended these ideas to graph data, they often rely on heuristic assumptions or lack theoretical guarantees."

---

> > > > > ### Author Response · Authors · 2025-04-21
> > > > > **Thank you for your comment**
> > > > >
> > > > > Thank you for your feedback! We have removed the duplicated sentence in Section 1 of the newly uploaded paper.

---

### Review · Reviewer_7C5j · 2025-03-06

**Summary Of Contributions:**

This paper proposes LR-GCL, a robust graph neural network (GNN) encoder designed for transductive node classification. LR-GCL leverages low-rank graph contrastive learning to handle noisy real-world graph data. It employs a two-step method—training a low-rank encoder with prototypical contrastive learning followed by a linear classifier—achieving strong performance and resilience. Theoretical generalization bounds and extensive experiments confirm its effectiveness.

**Audience:**

Yes

**Claims And Evidence:**

Yes

**Requested Changes:**

To further improve this paper, I suggest the authors make the following changes.

1. How the proposed LR-GCL is different from other related approaches should be clearly discussed in the manuscript.
2. The authors may consider improving the proposed LR-GCL to enhance the contributions of the paper.
3. The theoretical analysis can be presented in a rigorous form and with more explanations.
4. Experimental settings should be clearly introduced in the paper. Experimental results should be analyzed and discussed concretely.

**Strengths And Weaknesses:**

Strength
1. This paper is easy to follow.
2. Theoretical analysis makes this paper more solid.
3. The proposed method is evaluated in a lot of widely used datasets to show its effectiveness.

Weakness
1. The novelty is limited. Spectral GNNs can already denoise noise on features with approximation of spectral graph filters (e.g., SGC), while noisy edges have been explored by [1]. Besides, the frequency of graph signals has already been explored in many spectral GNNs.
2. The method of this paper is just a combination of existing methods in GNNs, including infoNCE, low-rank, clustering objective, etc. Currently, I do not see how the low-rank tech is related to the generalization bound.
3. The only interesting and novel points in the paper are the theoretical analysis of the generalization analysis of transductive learning. I strongly suggest that the authors include more explanations of this theory. Besides, the notation in the equation is not well explained.
4. In the experimental part, the settings of the experiments are not clear. For example, did the author use the same seed for 10 runs or use random seeds? Usually, semi-supervised learning and noise produce large stds. I wonder why the stds are very small.

[1] C. Zheng, B. Zong, W. Cheng, D. Song, J. Ni, W. Yu, H. Chen, and W. Wang, “Robust graph representation learning via neural sparsification,” in Proceedings of the 37th International Conference on Machine Learning, ICML, vol. 119, 2020, pp. 11 458–11 468.

---

> ### Author Response · Authors · 2025-04-12
> **Response to Reviewer 7C5j Part 1**
>
> We appreciate the review and the suggestions in this review. The raised issues are addressed below.
>
> **Responses to the Weakness**
>
> **1. “The novelty is limited...the frequency of graph signals has already been explored in many spectral GNNs.”**
>
> Although GNNs are considered low-pass filtering, they implicitly learn the low-frequency information, and the effect of such low-pass filtering is not strong enough to capture the Low-Frequency Property (LFP) in the noisy labels.  As illustrated by Figure 1 and Figure 3 in our papers, which demonstrate the LFP,  the majority of the clean label information is contained only in the low-rank part of the observed label (please refer to our response to weakness 4).
>
> In contrast with existing GNNs, our LR-GCL better captures the LFP in the noisy labels by learning low-rank features. We remark that low-rank learning exhibits superior performance for noisy attributes in [5] through learnable low-rank filters. Moreover, recent works on graph attention/transformer have shown that finding a good balance between low-frequency and high-frequency information in the graph benefits node representation learning for graph learning tasks such as node classification [3, 4]. Compared with the existing GNNs and graph attention/transformer methods, our LR-GCL learns a better balance between low-frequency and high-frequency information, with more focus on the low-frequency part by minimizing the truncated nuclear norm due to LFP. As shown in the new Table 3 of our revised paper, LR-GCL exhibits better node classification accuracy than graph attention/transformer methods, GFSA [3] and HONGAT [4], when label noise or attribute noise is present in the input graph. In addition, the balance between the low-frequency and high-frequency information can be quantitatively measured by the kernel complexity defined in Section 5.5. As shown in Table 5 and Table 6 of the revised paper, the node representations learned by LR-GCL exhibit lower kernel complexity than those of graph contrastive learning methods and graph attention/transformer methods.
>
>
> In summary, the major novelty and difference of our LR-GCL is that the low-rank learning is explicitly enforced in the training process of the GNN encoder by adding the Truncated Nuclear Norm (TNN) of the gram matrix of the node features to the training loss (Equation  (2)) of the LR-GCL encoder.  By optimizing such training loss of the LR-GCL encoder, our LR-GCL exhibits much lower Kernel Complexity (KC) compared to the competing methods, as shown in Table 5 and Table 6 of the revised paper. We also provided a theoretical guarantee in Theorem 4.1 on the upper bound for the test loss of a linear classifier trained on the node representations. A lower TNN leads to a lower KC, contributing to a lower upper bound for the test loss, which theoretically justifies the low-rank learning by optimization of the TNN. **Furthermore, inspired by the generalization bound for the test loss in Theorem 4.1, we propose LRA-LR-GCL, which further reduces the kernel complexity of LR-GCL in the revised paper**. Extended experimental results in Table 3 and Table 8 of the revised paper show that LRA-LR-GCL even outperforms LR-GCL with noisy labels and noisy attributes, and Table 5 and Table 6 of the revised paper show that LRA-LR-GCL exhibits lower KC and lower upper bound for the test loss than that of LR-GCL.

---

> > ### Author Response · Authors · 2025-04-12
> > **Response to Reviewer 7C5j Part 2**
> >
> > **2. “The method of this paper is just a combination of existing methods in GNNs, including infoNCE, low-rank...how the low-rank tech is related to the generalization bound.”**
> >
> > **3. “...I strongly suggest that the authors include more explanations of this theory...”**
> >
> > We have improved the presentation of our theoretical result in Theorem 4.1 of the revised paper. Equation (4)  in Theorem 4.1 of the revised paper shows that the upper bound for the test loss of a linear classifier trained on the node representations, or the generalization bound, for transductive node classification comprises the following three quantities, $L\_1(\mathbf{K}, \mathbf{\tilde{Y}}, t)$, $L\_2(\mathbf{K}, \mathbf{N}, t)$, and $\text{KC}({\mathbf{K}})$, which are explained as follows and also in the revised paper. $\mathbf{K}$ is the kernel gram matrix of the node representations $\mathbf{H}$ generated by the LR-GCL encoder. $\mathbf{\tilde{Y}}$ is the ground-truth clean label. $t$ is the epoch number. $\mathbf{N}$ is the label noise.
> >
> > - $L\_1(\mathbf{K}, \mathbf{\tilde{Y}}, t)$ corresponds to the training loss of the node classifier with the clean label.
> >
> > - $L\_2(\mathbf{K}, \mathbf{N}, t)$ corresponds to the loss incurred by the label noise.
> >
> > - $\text{KC}({\mathbf{K}})$ is the kernel complexity (KC), which measures the complexity of the kernel gram matrix $\mathbf{K} =\mathbf{H} \mathbf{H}^{\top}$ computed by the node representation $\mathbf H$.
> >
> > The KC $\text{KC}({\mathbf{K}})$ is computed using the truncated nuclear norm of the gram matrix of the node representations, $||\mathbf{K}||\_{r\_0}$, which is explicitly added to the training loss of our LR-GCL encoder in Equation (2). A lower $||\mathbf{K}||\_{r\_0}$ leads to a lower $\text{KC}({\mathbf{K}})$ contributing to a lower generalization bound, which justifies why we learn the low-rank features $\mathbf{H}$ in LR-GCL by adding the TNN $||\mathbf{K}||\_{r\_0}$ to the training loss of our LR-GCL encoder. A smaller $||\mathbf{K}||\_{r\_0}$ is obtained by optimizing such training loss, which in turn ensures a smaller KC contributing to a smaller generalization bound for transductive node classification, which is empirically observed in Table 5 and Table 6 of the revised paper.
> >
> > **4. “In the experimental part, the settings of the experiments are not clear...I wonder why the stds are very small.”**
> >
> > We generate the data with attribute noise or label noise ten times with different random seeds and calculate the mean and standard deviation of the node classification accuracy. The reason that our LR-GCL exhibits a lower standard deviation is attributed to the low-rank learning method in our LR-GCL. As illustrated by the LFP in Figure 2, Figure 3, and Figure 4 of the revised paper, the majority of the clean label information is contained only in the low-rank part of the observed label. This is also supported by the signal concentration ratio results in the caption of these three figures. In addition, the label noise is roughly distributed uniformly across the eigenvectors of the kernel gram matrix. As a result, the low-rank part of the observed noisy label contains the majority of the clean label information with only a small amount of the label noise.
> > The LFP and the signal concentration ratio results suggest that we can learn a low-rank portion of the observed noisy label $\mathbf Y$, which covers most information in the ground truth clean label while only learning a small portion of the label noise. As a result, by enforcing low-rank learning in the training of LR-GCL, the noise is largely mitigated.

---

> ### Author Response · Authors · 2025-04-12
> **Response to Reviewer 7C5j Part 3**
>
> **Responses to the Requested Changes**
>
>
> **1. “How the proposed LR-GCL is different from other related approaches...“**
>
> Please refer to our response to weakness 1.
>
> **2. “...improving the proposed LR-GCL to enhance the contributions of the paper.“**
>
> To further improve the performance of LR-GCL, we introduce a novel method, LRA-LR-GCL, in Section 4.3 of this paper. LRA-LR-GCL is inspired by the generalization bound for the test loss in Theorem 4.1, and it further reduces the kernel complexity of LR-GCL. Experimental results in Table 3 and Table 8 of the revised paper show that LRA-LR-GCL outperforms LR-GCL for noisy label and noisy attributes.
>
> LRA-LR-GCL features a novel LR-Attention layer, or the LRA layer, which applies self-attention to the output of the LR-GCL encoder by $\mathbf F = \mathbf B \mathbf H$, where $\mathbf H \in \mathbb R^{N \times d}$ is the low-rank node representations produced by the LR-GCL encoder through the optimization of Eq. (2). $\mathbf F$ is the attention output and $\mathbf B \in \mathbb R^{N \times N}$ is our new attention matrix in the LRA layer. We recall that the kernel gram matrix of the node features $\mathbf H$ is
>  $\mathbf K = \mathbf H \mathbf H^{\top}$.
>
> The attention weight matrix $\mathbf B$ is set to $\mathbf B = \mathbf K/{\hat \lambda_1}$. The gram matrix $\mathbf K_{\mathbf F}$ of the node representations $\mathbf F \in \mathbb R^{N \times d}$ is then $\mathbf K_{\mathbf F} = \mathbf F \mathbf F^{\top} = \mathbf K^3/{\hat \lambda_1^2}$.
> Let  $(\lambda_i) _ {i=1}^N$ be the eigenvalues of $\mathbf K_{\mathbf F}$ with $\lambda_1 \ge \lambda_2 \ge ...\lambda_N \ge 0$, then we have $\lambda_i = \hat \lambda_i^3/{\hat \lambda_1^2}$ for every $i \in [n]$. Noting that
> $\lambda_i = \hat \lambda_i \cdot \hat \lambda_i^2/{\hat \lambda_1^2} \le \hat \lambda_i$ due to $\lambda_1 \ge \lambda_i$ for all $i \in [N]$, therefore, the LRA layer can reduce the kernel complexity of the kernel gram matrix $\mathbf K$, because the KC of $\mathbf K_{\mathbf F}$ is always not greater than that of $\mathbf K$. We then train a transductive classifier on top of $\mathbf F$ similar to Section 4.2 by minimizing the following loss function
>
> $$\min\limits_{\mathbf W} L(\mathbf W) =  \frac{1}{m} \sum_{v_i \in \mathcal{V_L}} \textup{KL} ( \mathbf y_i, [\textup{softmax} (\mathbf F \mathbf W)]_i ),$$
>
> where $\mathbf W$ is the weight matrix for the classifier. Such a linear classifier trained with the LRA layer through the optimization of the above optimization problem is termed LRA-LR-GCL. It then follows from the above discussion and
> the upper bound for the test loss Eq. (4) in Theorem 4.1 that LRA-LR-GCL has a lower KC so that the test loss $\mathcal U_{\textup{test}}(t)$ of LRA-LR-GCL can be even lower than that of LR-GCL, suggesting a better prediction accuracy of LRA-LR-GCL than LR-GCL. This is empirically justified in Table 5 and Table 6, where LRA-LR-GCL exhibits lower KC and lower upper bound for the test loss than that of LR-GCL.
>
> **3. “The theoretical analysis can be presented in a rigorous form and with more explanations.“**
>
> Please refer to our response to weakness 3. We have improved the presentation of our theoretical result in Theorem 4.1 in the revised paper. Now the generalization bound for transductive node classification in Theorem 4.1 comprises the following three quantities, $L_1(\mathbf{K}, \mathbf{\tilde{Y}}, t)$, $L_2(\mathbf{K}, \mathbf{N}, t)$, and $\text{KC}({\mathbf{K}})$ with well-defined physical meanings.
>
> **4. “Experimental settings should be clearly introduced...“**
>
> Please refer to our response to weakness 4. We have added more details about the experimental settings in Section 5.1 of the revised paper.
>
>
>
> **References**
>
> [1] Wu, Felix, et al. "Simplifying graph convolutional networks." International conference on machine learning. 2019.
>
> [2] Dong, Yushun, et al. "Graph Neural Networks Are More Than Filters: Revisiting and Benchmarking from A Spectral Perspective." ICLR 2025.
>
> [3] Choi, Jeongwhan, et al. "Graph convolutions enrich the self-attention in transformers!." Advances in Neural Information Processing Systems 37 (2024): 52891-52936.
>
> [4] Zhang, Heng-Kai, et al. "HONGAT: graph attention networks in the presence of high-order neighbors." Proceedings of the AAAI Conference on Artificial Intelligence. Vol. 38. No. 15. 2024.
>
> [5] Cheng, Xiuyuan, et al. "Graph Convolution with Low-rank Learnable Local Filters." International Conference on Learning Representations. 2021.

---

> > ### Comment · Reviewer_7C5j · 2025-04-29
> >
> > Dear Authors,
> >
> > Thanks very much for your responses. All of my concerns have been addressed.

---

### Review · Reviewer_kMhi · 2025-04-06

**Summary Of Contributions:**

The paper proposes Low-Rank Graph Contrastive Learning (LR-GCL), a novel framework designed to enhance the robustness of graph neural networks (GNNs) against noise in node classification tasks. This work has several main contributions: 1) A low-rank regularized graph contrastive learning framework (LR-GCL) is proposed. By truncating the nuclear norm to constrain the low rank of node representation, the robustness and the generalization capabilities of node representations for GNNs is improved. 2) LR-GCL produces low-rank features inspired by the low frequency property of universal graph datasets and the sharp generalization bound for transductive learning. 3) The performance of LR-GCL is verified on 8 public dataset, which is better than the existing methods.

**Audience:**

Yes

**Broader Impact Concerns:**

No broader impact concerns needed.

**Claims And Evidence:**

Yes

**Requested Changes:**

Compare with low-rank GNN variants to isolate the performance improvement of LR-GCL comes from low-rank regularization, contrastive learning or the combination of both.
Optional Changes:
1) Minor writing issues: The introduction chapter jumps directly to 2 Related Works after 1.1 Contributions, which may make readers feel abrupt about the logical connection of subsequent content.
2) Discuss the computational cost of nuclear norm regularization and scalability to large graphs (e.g., training time/memory vs. graph size).

**Strengths And Weaknesses:**

# Strengths
1)  The integration of low-rank regularization into GCL is innovative, addressing a critical gap in noise-robust graph representation learning.
2) Provides a novel generalization bound for the test loss on the unlabeled data, and exhibits the advantage of learning with low-rank features for transductive classification with the presence of noise. The generalization bound and low-frequency property analysis provide a solid foundation for the method.
3) Experiments span diverse noise types (label/attribute, symmetric/asymmetric) and datasets, with ablation studies, and kernel complexity analysis to validate the approach’s effectiveness.
4) The proposed method is scalable and applicable to large graphs (e.g., ogbn-arxiv with 169k nodes).

# Weaknesses
1) The paper lacks verification of the independent role of low-rank regularization. Although LR-GCL is compared with semi-supervised node representation learning methods, node classification with label noise methods, GCL methods and contrastive learning baselines, the paper does not compare it with low-rank GNN variants. Therefore, it is unclear whether the performance improvement of LR-GCL comes from low-rank regularization itself or the combination of contrastive learning and low-rank regularization.
2) This work only focuses on the effect of Low-Rank Graph Contrastive Learning on transductive classification with the presence of noise, and the extension of inductive scenarios is the potential direction.

---

> ### Author Response · Authors · 2025-04-12
> **Response to Reviewer kMhi Part 1**
>
> We appreciate the review and the suggestions in this review. The raised issues are addressed below.
>
> **1. “ ... the paper does not compare it with low-rank GNN variants. Therefore, it is unclear whether the performance improvement of LR-GCL comes from low-rank regularization itself or the combination of contrastive learning and low-rank regularization.”**.
>
> To isolate and evaluate the effectiveness of the low-rank regularization independently of contrastive learning, we have conducted an ablation study where a semi-supervised GCN is trained with low-rank regularization, termed LR-GCN.
> The study is performed on Cora and Citeseer with asymmetric label noise, symmetric label noise, and attribute noise. The noise level is set to $60$% for all the three different types of noise in this ablation study.
>
> We first pre-train the GCN encoder in the transductive setting by minimizing the cross-entropy loss on the labeled nodes. Then, we train a transductive classifier on top of the node representations generated by the pre-trained GCN encoder by minimizing a joint objective that includes both the cross-entropy loss on labeled nodes and the truncated nuclear norm of the kernel gram matrix of the node representations.
> The results are shown in the table below. We also include the results of GCN and GCL for comparison, which are trained without the low-rank regularization. It is observed that the LR-GCN outperforms the GCN baseline, indicating that low-rank regularization alone is beneficial for the transductive node classification with the presence of noise. In addition, the GCL outperforms the GCN baseline, indicating that contrastive learning can also improve the robustness of GNNs against noise. Furthermore, LR-GCL, which combines both low-rank regularization and contrastive learning, achieves the best performance among all the methods.
>
> |Dataset|Noise Type|GCN|LR-GCN|GCL|LR-GCL|
> |:-:|:-:|:-:|:-:|:-:|:-:|
> |Cora|Asymmetric|0.405|0.442|0.470|0.492|
> |Cora|Symmetric|0.517|0.540|0.556|0.587|
> |Cora|Attribute|0.439|0.461|0.458|0.477|
> |Citeseer|Asymmetric|0.351|0.388|0.390|0.410|
> |Citeseer|Symmetric|0.341|0.387|0.425|0.455|
> |Citeseer|Attribute|0.372|0.395|0.390|0.406|
>
> **2. “ This work only focuses on the effect of Low-Rank Graph Contrastive Learning on transductive classification with the presence of noise, and the extension of inductive scenarios is the potential direction.”**
>
> To evaluate the effectiveness of the LR-GCL for inductive node classification with the presence of noise, we have conducted inductive node classification experiments following the inductive setting in [1]. The experiment is conducted on the Reddit dataset using the standard train/test split following [2], because Reddit contains more training nodes and a higher average degree compared to other datasets, making it better suited for evaluating the generalization capability of GNNs in the inductive node classification setting [2] . A three-layer GraphSAGE-GCN [2] with residual connections is employed as the inductive node encoder. We adopt the subsampling method proposed in [2], wherein a minibatch of nodes is first randomly selected at each training iteration. For each selected node, a corresponding subgraph is constructed by sampling its neighbors. We randomly sample 30, 25, and 20 neighbors at the first-, second-, and third-hop levels, respectively. As a result, the representation of each node is generated from a sampled subgraph. Next, we train the GraphSAGE-GCN encoder as our LR-GCL encoder in the inductive setting on the training nodes by minimizing the loss function in Equation (2) in Section 4.1 of our paper. Then the inductive LR-GCL classifier is trained by minimizing the regular cross-entropy on the labeled nodes in Equation (3) of Section 4.2 and use the trained $\mathbf W$ to predict the class labels of the test nodes by $\textup{softmax}(\mathbf h_i^{\top} \mathbf W)$ where $\mathbf h_i$ is the feature of each test node $i$ generated by the GraphSAGE-GCN encoder. The results are shown in the table below. We also include the results of the inductive GNN method, GraphSAGE-GCN [2], and the inductive GCL method, GRACE [1],  for comparison. It is observed that the LR-GCL outperforms all the baselines, indicating that low-rank regularization is beneficial for inductive node classification with the presence of noise.
>
> |Noise Type|Noise Level| GraphSAGE-GCN |GRACE |LR-GCL|
> |:-:|:-:|:-:|:-:|:-:|
> |Clean|0%| 93.0| 94.2| 94.5|
> |Asymmetric|40%| 71.9| 73.2| 75.6|
> |Asymmetric|60%| 54.9| 55.5| 57.2|
> |Asymmetric|80%| 38.0| 39.3| 41.7|
> |Symmetric|40%| 74.0| 75.2| 77.1|
> |Symmetric|60%| 51.3| 52.0| 53.6|
> |Symmetric|80%| 37.4| 38.3| 39.9|
> |Attribute|40%| 72.5| 72.9| 75.0|
> |Attribute|60%| 57.4| 58.9| 61.1|
> |Attribute|80%| 42.7| 42.4| 44.3|

---

> > ### Author Response · Authors · 2025-04-12
> > **Response to Reviewer kMhi Part 2**
> >
> > **3. “ Minor writing issues.”**
> >
> > We have added a brief description of the structure of the paper after Section 1.1 and before Section 2.  In Section 2, we review the existing graph neural networks, contrastive learning approaches, and robust learning techniques that motivate our method. Section 3 formally defines the learning objective, the notations, and the assumptions of our node classification task under noisy conditions. In Section 4, we present the formulation of the proposed Low-Rank Graph Contrastive Learning (LR-GCL) method with theoretical guarantee. Next, Section 5 validates our approach through extensive comparisons across benchmarks under varying noise conditions, demonstrating the superiority of LR-GCL.
> >
> >
> > **4. “Discuss the computational cost of nuclear norm regularization and scalability to large graphs (e.g., training time/memory vs. graph size).”**
> >
> > We have performed a comparison of the training time and memory consumption between LR-GCL and the ablation model GCL, which is trained without low-rank regularization. The comparison is conducted on three datasets, Cora, PubMed, and ogbn-arxiv, which have different sizes. The training on ogbn-arxiv is performed with a batch size of 16384. The training time is measured in seconds per epoch, and the memory consumption is measured in GB. The evaluation is performed on one Nvidia A100 GPU. We also include the classification accuracy with $60\%$ asymmetric label noise for LR-GCL and the GCL baseline. The results are shown in the table below. It is observed that although the LR-GCL has a slightly higher training time and memory consumption than the GCL baseline, it achieves significantly better classification accuracy under the presence of noise.
> >
> > | Dataset          | # Nodes   | # Edges    |Training Time (GCL) | Memory (GCL) | Accuracy (GCL)   | Training Time (LR-GCL) | Memory (LR-GCL) | Accuracy (LR-GCL)   |
> > |:-:|:-:|:-:|:-:|:-:|:-:|:-:|:-:|:-:|
> > | Cora             | 2,708   | 5,429     | 5.2    | 2.1       | 0.470 | 6.9   | 2.4       | 0.492 |
> > | PubMed           | 19,717  | 44,338    | 11.4      | 4.7       | 0.457 | 14.8     | 5.2      | 0.479 |
> > | ogbn-arxiv       | 169,343 | 1,166,243 | 25.7      | 7.5      | 0.379 | 31.9     | 9.4      | 0.405 |
> >
> >
> >
> > **References**
> >
> > [1] Zhu, Yanqiao, et al. "Deep graph contrastive representation learning." ICML Workshop on Graph Representation Learning and Beyond 2020.
> >
> > [2] Hamilton, Will, et al. "Inductive representation learning on large graphs." Advances in neural information processing systems 30 (2017).

---

> > > ### Comment · Reviewer_kMhi · 2025-05-20
> > >
> > > Dear Authors,
> > >
> > > Thanks very much for your reply. All of my concerns have been addressed.

---

### Note · Authors · 2025-06-21

I have read and agree with the venue's withdrawal policy on behalf of myself and my co-authors.

---

> ### Comment · Editors_In_Chief · 2025-06-25
>
> At 21 Jun 2025, 13:46 EDT, the EiCs modified the Decision from "Accept as is" to a "Reject" for the reasons outlined in the corresponding comment. Seven minutes later, 21 Jun 2025, 13:53 EDT, the authors attempted to withdraw the paper. Generally, rejected papers are not able to be withdrawn, the only reason this was possible is due to a technical delay in the system, in changing the status of the paper on OpenReview to Rejected (which was eventually done at 23 Jun 2025, 9:57 EDT). Thus the paper is not considered withdrawn, and this comment is kept only for posterity.

---

### Decision · Action_Editor_hgQH · 2025-06-04

**Recommendation:** Reject

**Comment:**

_Update by EiCs:_ A slightly modified version of this paper was concurrently submitted to the NeurIPS 2025 conference, in violation of the policies of the Transactions on Machine Learning Research (TMLR) on dual submissions and originality. As a result, this paper is rejected from TMLR, and the authors are prohibited from submitting to TMLR for a period of one year starting June 21, 2025.

-----
Original comment:

This paper empirically reveals the low-frequency nature of ground truth labels in graph data. Building on this insight, the authors introduce Low-Rank Graph Contrastive Learning (LR-GCL), which enhances standard prototypical graph contrastive learning by incorporating a truncated nuclear norm as a low-rank regularization term in the loss function. Both theoretical analysis and extensive experiments on public benchmarks validate LR-GCL’s effectiveness, demonstrating its superior performance and robustness in learning node representations.

In the revision, the authors have addressed the reviewers' concerns by incorporating additional experiments, expanding discussions and explanations. These revisions strengthen the paper's rigor and clarity, ensuring a more comprehensive and reliable contribution to the field.

**Audience:**

Update: See EiC comment below

-----

Original comment: Yes

**Claims And Evidence:**

Update: See EiC comment below

-----

Original comment: Yes